# Molecular and pharmacological evaluation of rare, cystic fibrosis-causing missense mutations of the CFTR channel

Olivér Závoti[1,2] 🆔 and László Csanády[1,2,3] 🆔

[1] *Department of Biochemistry, Semmelweis University, Budapest, Hungary*
[2] *HCEMM-SE, Molecular Channelopathies Research Group, Budapest, Hungary*
[3] *HUN-REN-SE, Ion Channel Research Group, Budapest, Hungary*

Handling Editors: Peying Fong & Tzyh-Chang Hwang

The peer review history is available in the Supporting information section of this article (https://doi.org/10.1113/JP288955#support-information-section).

**Abstract figure legend** Molecular pathomechanisms of five rare cystic fibrosis-associated mutations of the cystic fibrosis transmembrane conductance regulator (CFTR) anion channel. Top right, mutation loci mapped on the structure of the CFTR protein (pdbid: 6msm). Bottom left, Class II mutations cause defective processing of the channel protein, causing retention of the mutant protein in the endoplasmic reticulum and a decreased number ($N$) of channels on the cell surface. Class III mutations impair pore gating, resulting in a smaller open probability ($P_o$). Class IV mutations slow anion permeation through the open pore, diminishing the unitary current amplitude ($i$). The potentiator drugs ivacaftor and elexacaftor boost gating and the corrector drugs elexacaftor and tezacaftor enhance maturation for all five CFTR variants.

**Abstract** Cystic fibrosis (CF) is a life-threatening condition caused by pathogenic mutations of the cystic fibrosis transmembrane conductance regulator (CFTR) anion channel. While most patients with frequent mutations are treated with a combination of corrector and potentiator drugs (elexacaftor–tezacaftor–ivacaftor, ETI), for many rare variants no such treatment is approved. Here we characterized the molecular pathologies of five rare CF-associated mutants (G126D, I336K, T465I, T582I and D984V), and evaluated their responsiveness to ETI treatment. Channel expression was investigated by western blot from HEK-293T cells transiently expressing mutant channels, with or without corrector drugs (elexacaftor and tezacaftor). Drug efficiency was assessed by the enhancement of channel glycosylation. Maturation was mildly (T582I, G126D) to severely (T465I) defective for all five mutants, but could be restored to wild-type levels by incubation with corrector drugs. Gating properties and responses to potentiator drugs (ivacaftor and elexacaftor) were assessed in macroscopic and single-channel inside-out patch-clamp recordings from mutant channels expressed in *Xenopus laevis* oocytes. Mutations G126D, T582I, I336K and T465I markedly decreased channel open probability, with the greatest reduction being caused by mutation T465I. Mutations G126D, I336K and T465I also slightly impaired unitary conductance. Applying potentiator drugs boosted gating for all variants, producing multifold increases in macroscopic currents. In conclusion, all five mutations impair channel maturation and gating to various degrees, but a normal glycosylation pattern can be restored with correctors, and channel gating can be enhanced with potentiators. These *in vitro* observations suggest that ETI treatment would be beneficial for CF patients carrying an allele with any of the five mutations.

(Received 25 March 2025; accepted after revision 2 July 2025; first published online 7 August 2025)

**Corresponding author** László Csanády: Department of Biochemistry, Semmelweis University, Budapest, Hungary. Email: csanady.laszlo@semmelweis.hu

## Key points

- Mutations of the cystic fibrosis transmembrane conductance regulator (CFTR) anion channel cause cystic fibrosis.
- Highly efficient modulator therapy has been approved for ∼90% of cystic fibrosis patients, but a large number of rare mutations are not yet eligible for treatment.
- Five rare CFTR mutations found in cystic fibrosis patients in Hungary and worldwide are shown here to produce complex channel pathologies that to differing degrees affect protein maturation, channel gating and anion permeation through the open pore.
- All five variants respond strongly to clinically employed modulator drugs that boost channel surface expression and gating.
- The results help us understand how these CFTR mutations lead to cystic fibrosis, and suggest that patients carrying any of the five variants would probably benefit from modulator therapy.

**Olivér Závoti** obtained his medical degree in 2023 and is currently pursuing PhD studies in cellular and molecular physiology at Semmelweis University, Budapest. His research focuses on the CFTR (cystic fibrosis transmembrane conductance regulator) anion channel, mutations of which are responsible for the development of cystic fibrosis. He has been investigating the functional consequences of multiple rare pathogenic CFTR variants and their potential pharmacological modulation. In the course of his work, he has gained expertise in patch-clamp electrophysiology, molecular biology, cell culture and protein biochemistry.

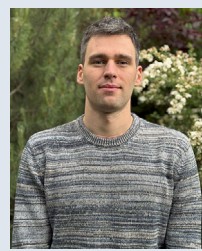

# Introduction

The cystic fibrosis transmembrane conductance regulator (CFTR) anion channel (Anderson et al., 1991; Bear et al., 1992) is located on the apical surface of epithelial cells and plays a role in salt-water transport (Quinton, 1990). It is an ATP-binding cassette (ABC) protein (ABCC7) and exhibits the typical structural elements of the ABC protein superfamily. The channel is a monomer, and contains two transmembrane domains (TMD1, TMD2), two cytosolic nucleotide binding domains (NBD1, NBD2) and a unique regulatory (R) domain (Riordan et al., 1989). Channel gating shows a bursting pattern, and is fuelled by the binding and hydrolysis of ATP at the NBDs (Csanády et al., 2010; Gunderson & Kopito, 1995). ATP binding to both NBDs drives formation of a tight head-to-tail NBD dimer which traps two ATP molecules at the dimer interface, in composite binding sites (Sites 1 and 2) formed by the Walker A and B motifs of one NBD and the ABC signature sequence of the other (Zhang et al., 2018). ATP hydrolysis at the site flanked by the Walker motifs of NBD2 (Site 2) results in loosening of the dimer interface allowing nucleotide exchange at this site, whereas Site 1, flanked by the Walker motifs of NBD1, is catalytically inactive (Aleksandrov et al., 2002; Basso et al., 2003) and remains dimerized throughout multiple ATPase cycles of Site 2 (Levring et al., 2023; Szollosi et al., 2011; Tsai et al., 2010). These NBD conformational changes drive pore gating: initiation of an open burst is linked to tightening, and termination of the burst to loosening, of the dimer interface around Site 2 (Vergani et al., 2005). Phosphorylation of the R domain by the protein kinase A catalytic subunit (PKA) is a prerequisite for channel activity (Cheng et al., 1991). The unphosphorylated R domain is wedged between the two NBDs and prevents their dimerization (Liu et al., 2017), and upon phosphorylation the R domain is released from its inhibitory position allowing channel gating (Zhang et al., 2018). In addition, simple binding of PKA to CFTR causes additional, reversible stimulation of channel open probability (Fiedorczuk et al., 2024; Mihályi et al., 2020, 2024).

Pathogenic mutations of the CFTR channel cause the disease cystic fibrosis (CF), the most common lethal genetic disorder in the Caucasian population which affects ∼100,000 patients worldwide (Bell et al., 2020; de Boeck & Amaral, 2016; O'Sullivan & Freedman, 2009). CF affects multiple organs, predominantly the lung, intestine, pancreas and liver, causing an imbalance in electrolyte homeostasis (Mall et al., 2024). Based on the molecular pathomechanism, CF-associated CFTR mutations have been classified into six categories (Haardt et al., 1999; Welsh & Smith, 1993; Zielenski & Tsui, 1995), but most mutations belong to Classes I–IV. Class I mutations are nonsense mutations, resulting in the absence of functional protein. Class II mutations impair proper folding and maturation, causing retention of the channels in the endoplasmic reticulum (ER) followed by degradation in the proteasome. Class III comprises mutant channels with defective gating, while Class IV mutations impair anion permeation through the open pore. Although several hundred CF-causing mutations have been identified, the deletion of phenylalanine 508 (F508del), a Class II/III mutation which also shortens the residence time of the protein in the plasma membrane (Lukacs et al., 1993), accounts for ∼70% of all pathogenic alleles (O'Sullivan & Freedman, 2009).

The total CFTR current across the apical plasma membrane of an epithelial cell ($I_{CFTR}$) is the product of the number of channels expressed at the surface ($N$), the open probability ($P_o$), and the unitary current amplitude ($i$). Mutations that fall into functional Classes I–II, III and IV reduce $N$, $P_o$ and $i$, respectively. Although CF is incurable, pharmacotherapy using the CFTR potentiator drug ivacaftor (VX-770) to stimulate gating ($P_o$), and the corrector drugs tezacaftor (VX-661) and elexacaftor (VX-445) to boost channel surface expression ($N$), has shown remarkable clinical success. As of today, modulator therapy using a combination of elexacaftor, tezacaftor and ivacaftor (ETI, developed by Vertex Pharmaceuticals) has been approved by the U.S. Food and Drug Administration (FDA) for patients who carry mutation F508del (Barry et al., 2021; Hoy, 2019), or certain other rare variants, on at least one allele. However, for many rare variants, ETI treatment has not yet been approved. Characterizing the phenotypes and drug responses of such rare mutations is therefore a timely problem.

Recently, five rare CF-causing missense mutations [c.377G>A (p.G126D; legacy: G126D), c.1007T>A (p.I336K; legacy: I336K), c.1394C>T (p.T465I; legacy: T465I), c.1745C>T (p.T582I; legacy: T582I), c.2951A>T (p.D984V; legacy: D984V)] were identified in the Hungarian CF population (Deák et al., 2022). Although these mutations (except for D984V) had been previously reported in sequencing studies worldwide (Cheng et al., 2022; Cuppens et al., 1993; des Georges et al., 2004; Férec et al., 1993; Fustik et al., 2021; Hatton et al., 2022; Ivády et al., 2018; Krenková et al., 2009; Ockenga et al., 2000; Sobczynska-Tomaszewska et al., 2013; Wagner et al., 1994), their impacts on CFTR biogenesis and function and the responsiveness of the mutant channels to drugs have not yet been studied in detail. Here we aimed to investigate the functional consequences of each of these five mutations, and to evaluate the responsiveness of the mutant channels to available CFTR potentiator and corrector drugs.

## Methods

### Ethical approval

Animal experiments were approved by the Semmelweis University Animal Welfare Body (approval number: SEMMAWB/2023-001). Semmelweis University operates following the guidelines of the Hungarian Medical Research Council (Egészségügyi Tudományos Tanács). All animal research within this study adheres to the policies of *The Journal of Physiology* regarding animal experiments.

### Molecular biology

The mutations were introduced into the human CFTR/pGEMHE and CFTR/pcDNA3 sequences using the QuikChange II XL Site-Directed Mutagenesis Kit (Agilent Technologies, Santa Clara, CA, USA). Plasmid DNA was purified using the QIAGEN HiSpeed Plasmid Midi Kit, and the mutations were confirmed by automated sequencing of the entire cDNA insert (LGC Genomics GmbH, Berlin, Germany). The CFTR/pGEMHE plasmids were then linearized using the NheI HF restriction endonuclease (New England Biolabs, Ipswich, MA, USA) and transcribed in vitro using the mMessage-mMachine T7 Ultra Kit (Agilent Technologies). All purified cRNAs were stored at −80°C.

### Channel expression in *Xenopus laevis* oocytes

Adult female *Xenopus laevis* [RRID: NXR_0.0080] were kept at ∼18°C, under a 12 h/12 h light/dark cycle, and had free access to food. Animals were anaesthetized according to Institutional Animal Care and Use Committee guidelines by submersion into a tricaine (1%) solution, and oocytes were extracted by partial ovariectomy. Animals were killed by freezing (−20°C) before the return of consciousness. Oocytes were digested with Collagenase Type II. Separated oocytes were stored at 18°C in a solution containing 82 mM NaCl, 2 mM KCl, 1 mM MgCl₂, 5 mM Hepes (pH 7.5 with NaOH), supplemented with 1.8 mM CaCl₂ and 50 µg/ml gentamycin. Oocytes were injected with 0.1–30 ng cRNA, in a fixed volume of 50 nl, using a Nanoject II injector (Drummond Scientific, Broomall, PA, USA). Recordings were made 1–3 days after the injection.

### Inside-out patch-clamp recordings

The patch pipette solution contained 138 mM *N*-methyl-ᴅ-glucamine (NMDG), 2 mM MgCl₂, 5 mM Hepes, and pH was set to 7.4 with HCl. The bath solution contained 138 mM NMDG, 2 mM MgCl₂, 5 mM Hepes, 0.5 mM EGTA, and pH was set to 7.1 with HCl. Excised inside-out patches were positioned into a flow chamber in which the continuously flowing bath solutions could be exchanged with a time constant of <100 ms using ALA-VM8 electronic valves (Ala Scientific Instruments, Farmingdale, NY, USA). All recordings were performed at 25°C. Macroscopic currents were recorded at −40 mV, single-channel currents at −80 mV, and conductance measurements were obtained at membrane potentials ranging from −80 to +40 mV. Amplified currents were low-pass filtered at 1 kHz (Axopatch 200B, Molecular Devices, Sunnyvale, CA, USA), digitized at 10 kHz (Digidata 1550B, Molecular Devices) and recorded to disk (Pclamp 11, Molecular Devices [RRID: SCR_011323]). MgATP was diluted into the bath solution from a 400 mM aqueous stock (pH 7.1 with NMDG). The catalytic subunit of bovine protein kinase A (PKA) was prepared from beef heart as described (Mihályi et al., 2024). Vx-770 (ivacaftor) and Vx-445 (elexacaftor) were added from >100× stock solutions in DMSO, and 2′-deoxy-N6-(2-phenylethyl)adenosine-5′-*O*-triphosphate (P-dATP) was directly dissolved into the bath solution at a concentration of 50 µM. Each recording started with the following protocol: a 40–60 s baseline was recorded, then 2 mM ATP was added for 40–60 s, followed by a 1–3 min exposure to 300 nM PKA to phosphorylate and fully activate the channels. Apparent affinities for ATP, single-channel gating and potentiator responses were assessed following PKA washout.

### Kinetic analysis of the electrophysiological recordings

All raw traces were Gaussian filtered at 50 or 100 Hz, and the baseline was subtracted. Recordings with five or fewer active channels were idealized by half-amplitude threshold crossing. The events lists were fitted with the $C_1 \leftrightarrow O_3 \leftrightarrow C_2$ model using a simultaneous maximum likelihood fit to the dwell-time distributions at all conductance levels, with an imposed fixed dead time of 4 ms (Csanády, 2000). The obtained transition rate constants $k_{12}$, $k_{21}$, $k_{23}$ and $k_{32}$ were used to calculate mean burst ($T_b$) and interburst ($T_{ib}$) durations as $T_b = (1/k_{31})(1+k_{32}/k_{23})$, and $T_{ib} = 1/k_{13}$. Opening and closing rate were defined as the rate of entering ($1/T_{ib}$) and exiting ($1/T_b$) a burst, respectively.

For I336K and T465I, upper bounds for $P_o$ in 2 mM ATP were obtained from the maximal fractional stimulation by drugs (cf. Fig. 5*F*, *right*): $P_o \leq I_{ATP}/I_{max}$, where $I_{ATP}$ is the average current in 2 mM ATP, and $I_{max}$ is the maximal observed current in the presence of P-dATP + PKA + VX-770 + VX-445 in the same patch. Closing rate ($1/T_b$) in 2 mM ATP was determined from dwell-time analysis as for the other three constructs, and an upper bound for opening rate ($1/T_{ib}$) in 2 mM ATP was obtained from the relationship $P_o \approx T_b/(T_b+T_{ib})$. Thus, $(1/T_{ib}) \leq (1/T_b) \cdot I_{ATP}/(I_{max} - I_{ATP})$.

## Channel expression in HEK-293T cells

HEK 293T cells were cultured in T-175 flasks at 37°C in Dulbecco's Modified Eagle's Medium (DMEM) with 4.5 g/l glucose supplemented with 10% fetal bovine serum (FBS), 2 mM L-glutamine, and 100 units/ml penicillin/streptomycin. After reaching ∼70–80% cell confluency, the cell cultures were transiently transfected with the CFTR/pcDNA3 constructs using Lipofectamine 3000 Transfection Reagent (Thermo Fisher, Waltham, MA, USA) according to the protocol provided by the manufacturer.

## Crude membrane preparations

After 48 h of incubation following transfection, upon reaching ∼100% cell confluency, the cells were washed with ice-cold divalent-free phosphate-buffered saline (PBS), then 1.3 ml Lysis Buffer (10 mM Hepes, 1 mM EDTA, 1 mM phenylmethylsulfonyl fluoride (PMSF), 1 μg/ml leupeptin and 1 μg/ml pepstatin, pH 7.2) was added to each flask. The cells were scraped off the flasks and pipetted into a Dounce homogenizer. After a 5 min incubation on ice, the cells were ruptured with 10 strokes. Then, 325 μl sucrose buffer (1.25 M sucrose, 10 mM Hepes, pH 7.2) was added to the lysate and mixed with a single stroke. Cell debris was pelleted at 4°C and 3000 $g$ for 10 min. The supernatant was centrifuged at 4°C and 16,000 $g$ for 30 min. The pellet was resuspended with 50 μl of Sample Buffer (62.5 mM Tris-HCl, 3% SDS, 10% glycerol, 0.05% bromophenol-blue, pH 6.8). For the corrector drug experiments, the cell cultures were transfected at ∼50–60% cell confluency. Then, 24 h after transfection, 3 μM Vx-445 (elexacaftor) and 3 μM Vx-661 (tezacaftor) were added to the media. After an additional 24 h, the cells were ruptured and crude membrane preparations were made as described above.

## Western blot

Aliquots of 3 μl of the above-described crude membrane preparations were diluted with 7 μl of Sample Buffer. After adding 0.5 μl of 1 mM DTT, the samples were incubated at 65°C for 15 min, loaded onto 7.5% polyacrylamide gels, and separated at a voltage of 80–120 mV. The gels were transferred to nitrocellulose membranes by a 10 min run on a Bio-Rad Trans Blot Turbo system at 2.5 A and 25 V. Following transfer, the membranes were blocked for 1 h in 5% BSA dissolved in Tris-buffered saline (TBS) supplemented with 0.2% Tween 20 (TBST). Primary antibodies were provided by Dr. Martina Gentzsch's laboratory at the University of North Carolina at Chapel Hill through the Cystic Fibrosis Foundation. For our studies, the 5E2 monoclonal antibody, which recognizes the C-terminal portion of NBD2 (residues 1371–1385), was used at 1:5000 dilution (in TBST + 0.5% BSA). After overnight incubation with the primary antibody at room temperature, the membranes were washed 3× for 10 min in TBST + 0.5% BSA. Then, alkaline-phosphatase-conjugated anti-mouse secondary antibody was added at 1:10,000 dilution. After 2 h of incubation, the membranes were washed 3× for 10 min with TBST. The signal was detected by adding 10 ml alkaline phosphatase buffer, made according to the manufacturer's (Roche, Indianapolis, IN, USA) protocol, supplemented with 50 μl of 4-nitroblue tetrazolium chloride (NBT) and 37.5 μl of 5-bromo-4-chloro-3-indoyl phosphate (BCIP) solutions.

## Densitometric analysis

For densitometric analysis membranes were scanned as 8-bit greyscale images; all pixel densities remained well below 255. Within each lane, pixel densities were then averaged across rows of pixels, and plotted as a function of vertical position, resulting in densitograms for each lane. Following subtraction of the densitogram of the control lane, 'baseline-subtracted' densitograms were fitted with sums of three Gaussian functions, corresponding to the positions of Bands A, B and C. Band A, B and C densities were defined as the areas under the respective Gaussian component. Finally, Band C densities were normalized either to that of wild-type (WT) CFTR on the same blot (Fig. 2*C*), or to the sum of the densities of the three bands for the respective construct (Fig. 2*D*).

## Statistics

All data are given as mean ± SD. The number of electrophysiological experiments ('*n*') shown in the figures indicates the number of excised patches. For each independent biological replicate used for western blots, all six constructs were handled in parallel: the cell cultures were passaged, transfected and lysed simultaneously, on the same days. Significance levels were obtained using Student's *t* test. The threshold of significance was chosen to be $P < 0.05$.

## Reagents

For the molecular biology the QuikChange II XL Site-Directed Mutagenesis Kit was purchased from Agilent Technologies (Cat# 200522), the HiSpeed Plasmid Midi Kit from Qiagen (Valencia, CA, USA; Cat# 12643), the NheI HF restriction endonuclease from New England Biolabs (Cat# R3131), and the mMessage-mMachine T7 Ultra Kit from Agilent Technologies (Cat# 200522). Type II Collagenase was from Thermo Fisher Scientific (Cat# 17101015). For the patch-clamp experiments

MgATP was obtained from Merck (Darmstadt, Germany; Cat# A9187) and P-dATP from Biolog LSI (Cat# D 104–05). The CFTR modulators Vx-770 (ivacaftor), Vx-445 (elexacaftor) and Vx-661 (tezacaftor) were purchased from Selleckchem (Cat# S1144, S8851 and S7059, respectively). For the cell cultures, HEK 293T cells were obtained from ATCC (Rockville, MD, USA; Cat# CRL-11268), DMEM with 4.5 g/l glucose from Merck (Cat# D6429), FBS from EuroClone (Milan, Italy; Cat# ECS0180L), penicillin/streptomycin from Lonza (Basel, Switzerland; Cat# 17–602E), and the Lipofectamine 3000 Transfection Reagent from Thermo Fisher Scientific (Cat# L3000015). For the membrane preparations, divalent-free PBS was purchased from BioConcept (Allschwil, Switzerland; Cat# 3–05F29-I). For the western blots, BSA was obtained from Merck (Cat# A7906), the 5E2 primary CFTR antibody from the Antibody Distribution Program of the Cystic Fibrosis Foundation, the alkaline-phosphatase-conjugated anti-mouse secondary antibody from Merck (Cat# A4312 [RRID: AB_258154]), and NBT and BCIP solutions from Roche (Cat# 11383213001 and 11383221001, respectively).

## Results

### The five CF-causing mutations are scattered throughout the channel structure

Mapping the locations of the five CF-causing mutations onto the structure of phosphorylated, ATP-bound 'quasi-open' human CFTR (PDBID: 6msm; Zhang et al., 2018) illustrates how these positions are scattered throughout the protein (Fig. 1*A*). G126 (*red spacefill*) and I336 (*blue spacefill*) are located in transmembrane helices 2 and 6 (TM2 and TM6), respectively, near the extracellular membrane surface. D984 (*orange spacefill*) is situated in TM9, near the intracellular membrane surface, while T465 (*magenta spacefill*) and T582 (*cyan spacefill*) are both found in NBD1.

Closer inspection of these residues provides some hints for their potential involvement in channel function. G126 and I336 are positioned near the constriction that forms the selectivity filter (Levring & Chen, 2024), but neither residue is facing the channel pore (Fig. 1*B*, panels 1–3). However, I336 is immediately adjacent to residue F337, the side chain of which contributes to the coordination of dehydrated chloride ions as they permeate through the selectivity filter (Levring & Chen, 2024). Moreover, G126 is located at a helical kink in TM2 which was suggested, based on molecular dynamics simulations, to change conformation during gating (Zeng et al., 2025). T465 is part of the conserved Walker A motif of NBD1, and its side chain provides the closest coordination ligand for the $Mg^{2+}$ ion in degenerate Site 1 (Fig. 1*B*, panel

4). The T582 side chain forms an H-bond with the side chain of D579 in the NBD1 D-loop (Fig. 1*B*, panel 5), which contributes to the lining of Site 2 (Gao et al., 2024; Simon et al., 2023; Zhang et al., 2018). Finally, the D984 side chain is within reach to form a salt bridge with that of R289 in TM5 (Fig. 1*B*, panel 6). Of note, in the structure of dephosphorylated ATP-free closed

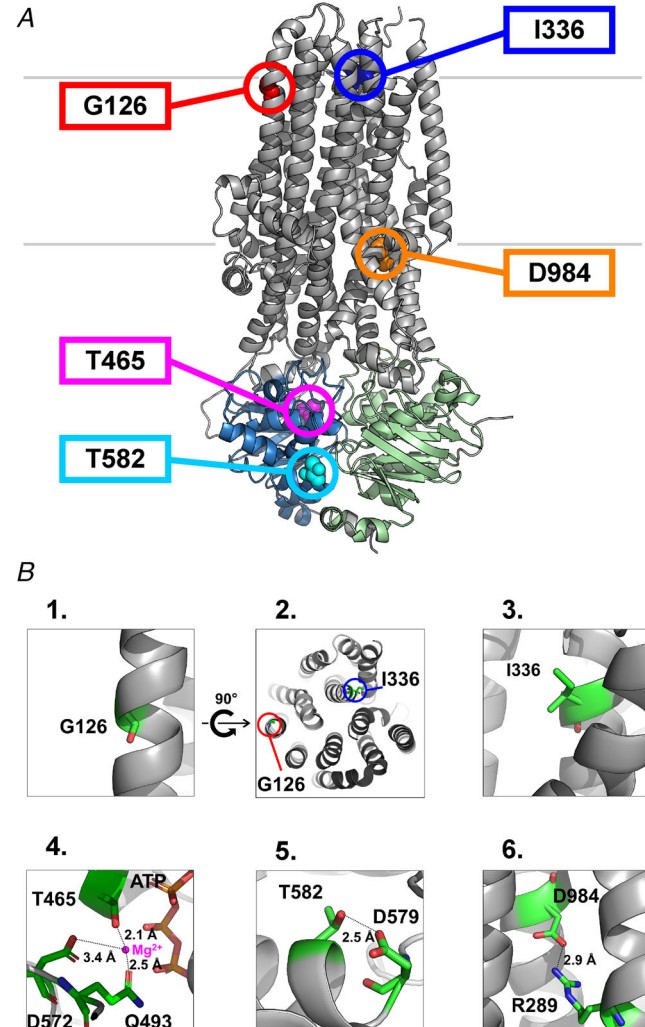

**Figure 1. Localization of the five target positions affected by rare CF-causing mutations**

*A*, the positions of the five CF-causing mutations mapped on the structure of phosphorylated, ATP-bound quasi-open human CFTR (PDB: 6msm). Colour coding: TMDs, *grey*; NBD1, *marine*; NBD2, *green*. Location of the target residues: G126 (*red spacefill*), transmembrane helix 2 (TM2); I336 (*blue spacefill*), TM6; T465 (*magenta spacefill*), NBD1; T582 (*cyan spacefill*), NBD1; D984 (*orange spacefill*), TM9. *B*, panels 1–6: close-up views of the target residues and their possible interactions. The G126 and the I336 side chains are not facing the channel pore. The T465 side chain is the closest of the three side chains that coordinate the $Mg^{2+}$ ion in the degenerate site. The T582 and D984 side chains might form an H-bond and a salt bridge, respectively, probably involved in stabilizing the open-pore structure.

human CFTR (PDBID: 5uak; Liu et al., 2017) side-chain densities for D579, T582 and D984 are absent, but for both position pairs 582–579 and 984–289 the alpha carbons move further apart from each other, suggesting that the above side chain interactions may weaken or not form in closed channels. Interestingly, mutation D579G is also associated with CF (Picci et al., 1999).

## Channel expression/maturation of the five mutant variants is mildly to severely compromised, but restored by the correctors VX-661 + VX-445

During normal maturation, the CFTR protein undergoes core glycosylation in the ER, followed by complex glycosylation in the Golgi. Because glycosylation reduces electrophoretic mobility, the various stages of this process can be resolved on SDS-PAGE gels which typically display three bands for WT CFTR (Fig. 2*A*, *left*), corresponding to the non-glycosylated (Band A), core-glycosylated (Band B), and mature fully glycosylated (Band C) form, respectively (Gregory et al., 1990). Class II mutations, including the most common CF-causing mutation F508del, impair protein maturation, causing retention of the protein in the ER and consequent loss of Band C (Cheng et al., 1990).

To assess potential effects of the five rare CF mutations on CFTR expression/maturation, we compared the electrophoretic migration patterns of the mutants on western blots of crude membrane preparations, from HEK-293T cells transiently transfected with either of the five constructs (Fig. 2*A*). Densitometric analysis revealed a decreased Band C intensity for all five mutants when compared to WT (Fig. 2*C*, *light grey bars*), reaching only ∼38% and ∼14% of WT, respectively, for I336K and T465I. Because the densities for Bands A and B did not decrease for any of the mutants, the fractional density of Band C [C/(A + B + C)] was also significantly reduced for I336K ($P = 0.0116$) and T465I ($P = 0.00005$) (Fig. 2*D*, *light grey bars*). Insofar as Band C intensity is a reflection of channel surface expression (Bihler et al., 2024; Cheng et al., 1990), these data suggest a decreased surface density for all mutant channels, but especially for I336K and T465I CFTR, classifying them as Class II mutants.

The corrector drug combination VX-661 + VX-445 (tezacaftor–elexacaftor) efficiently rescues maturation of Class II CFTR mutants, including F508del (Keating et al., 2018) and a large number of rare variants (Bihler et al., 2024). To assess the responsiveness to correctors of the five mutants studied here, cells were incubated for 24 h with 3 µM VX-661 + 3 µM VX-445 prior to harvesting (Bihler et al., 2024; Laselva et al., 2021); such drug concentrations are both physiologically relevant (Pigliasco et al., 2023; Ryan et al., 2022) and were found maximally effective on

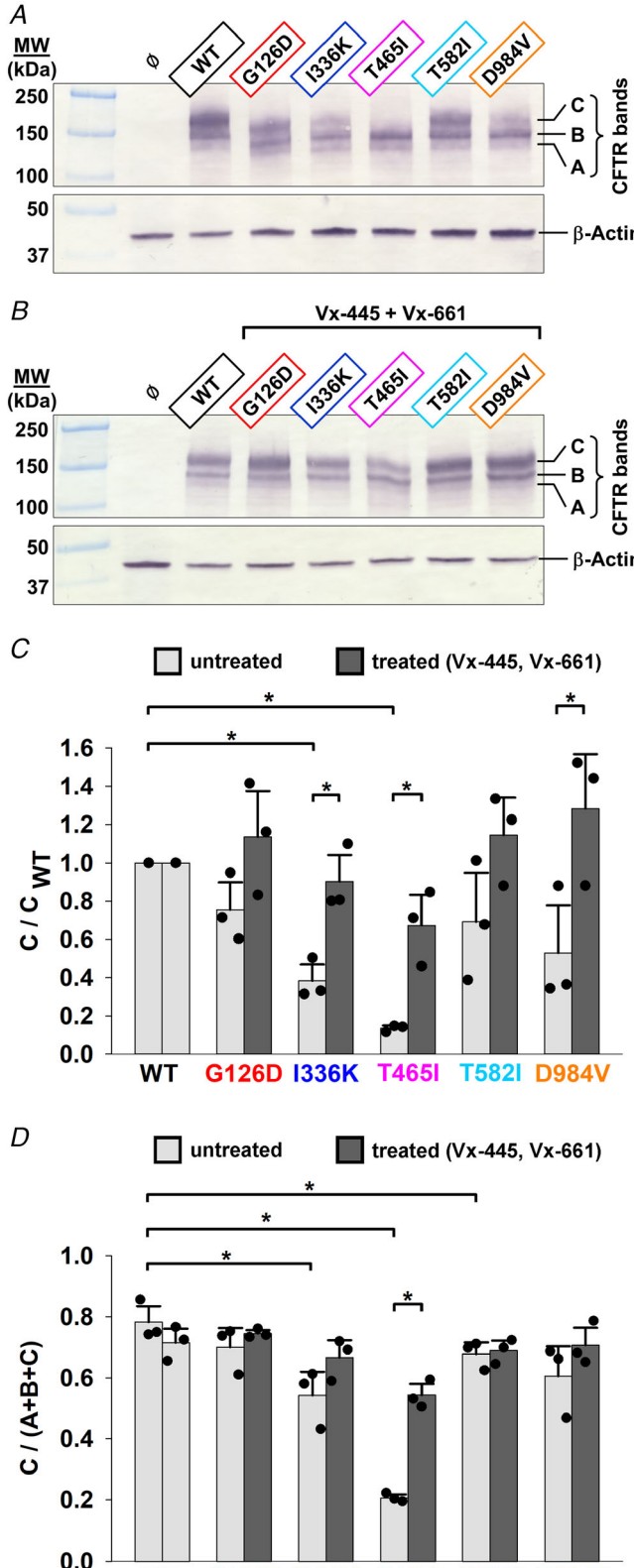

**Figure 2. Impaired maturation of mutant CFTR channels and rescue by CFTR corrector drugs**
*A* and *B*, western blot analysis of wild-type (WT) and mutant CFTR channel constructs from (*A*) non-treated HEK-293T cells, or (*B*) HEK-293T cells incubated in the presence of 3 µM Vx-445 + 3 µM

Vx-661. In both *A* and *B*, the WT sample represents non-treated cells, the negative control (*lane Ø*) non-transfected cells, and *β*-actin was used as a loading control (*bottom panels*). *C* and *D*, densitometric analysis of the data in *A* and *B* for each construct; the left column represents the data in *A* and the right column the data in *B*. Band *C* densities of CFTR constructs without (*light grey bars*, from *A*) and following (*dark grey bars*, from *B*) corrector treatment are shown normalized either (*C*) to the Band *C* density of untreated WT CFTR on the same blot, or (*D*) to the sum of the densities of Bands A, B and C of the respective construct. Bars represent mean ± SD from three biologically independent samples, and asterisks highlight statistically significant differences (*P* < 0.05).

multiple mutants (Phuan et al., 2021; Veit et al., 2020). Corrector treatment enhanced Band C density for all five variants (compare Fig. 2*A* with 2*B*), and the fractional drug response was inversely related to the Band C density of untreated cells, being largest for T465I and I336K (Fig. 2*C*–*D*, *dark* vs. *light grey bars*). Notably, following drug treatment, Band C density was restored to near WT levels for all mutants (Fig. 2*C*, *dark grey bars*).

### Regulation by PKA and sensitivity to ATP are intact for G126D, T582I and D984V CFTR

To assess mutational effects on channel function, WT CFTR and all five variants were expressed in *Xenopus laevis* oocytes, and studied in inside-out patch-clamp recordings. Whereas for the G126D, T582I and D984V mutants cytosolic superfusion with 300 nM PKA + 2 mM ATP routinely activated large macroscopic currents (Fig. 3*B*–*D*), for I336K and T465I only sporadic pore openings could be observed under identical conditions, even following injection of up to 30 ng of cRNA. Gating properties could therefore be quantitatively characterized only for G126D, T582I and D984V CFTR.

Similarly to WT CFTR (Fig. 3*A*), all three mutants (Fig. 3*B*–*D*) showed intact regulation by PKA: negligible activity in ATP prior to PKA exposure, robust activation by PKA and rapid partial deactivation upon PKA washout, the last reflecting loss of the reversible component of activation caused by PKA binding (Mihályi et al., 2020). Of note, the fractional current that survives PKA removal (Fig. 3*E*, *grey bars*) was slightly but significantly reduced for all three mutants compared to WT (*P* = 0.000597, 0.00009 and 0.0223 for G126D, T582I and D984V, respectively), suggesting a slightly increased fractional contribution of the reversible component to total channel activity in the mutants.

The sensitivities of pre-phosphorylated channels to ATP, assessed by exposures of patches with multiple channels to various test ATP concentrations bracketed by applications of 3.2 mM ATP (Fig. 3*F*–*I*), were intact for all three mutants, yielding dose–response curves indistinguishable from WT ($K_{1/2} \sim 50$ μM; Fig. 3*J*).

### Single-channel recordings reveal gating defects for G126D, T582I and D984V CFTR

To compare steady-state gating kinetics of the mutant channels to that of WT CFTR, we recorded currents of pre-phosphorylated WT, G126D, T582I and D984V channels in the presence of 2 mM ATP, from patches containing only small numbers of channels ($N \leq 5$) in which individual gating transitions could be clearly resolved (Fig. 4*A*, *left*). In an attempt to estimate the number of channels in the patch, at the end of each recording 300 nM PKA was reapplied to provide reversible stimulation, and ATP was replaced with 50 μM P-dATP, which stimulates CFTR gating by binding directly to CFTR's NBDs (Miki et al., 2010). This combination results in substantial gating stimulation (Sorum et al., 2017) (Fig. 4*A*, *right*). Open probabilities, mean open burst ($T_b$) and closed interburst ($T_{ib}$) durations were extracted by dwell-time analysis (Csanády, 2000). Compared to WT CFTR, $P_o$ was only slightly reduced for D984V, but significantly reduced for G126D (*P* = 0.00194) and T582I (*P* = 0.000195) (Fig. 4*B*, *right*), assigning the latter two mutations also to Class III.

The underlying changes in gating kinetics were clearly different for the various mutants: the observed reduction in $P_o$ was caused by an acceleration of closing rate (shortening of $T_b$) for G126D (*P* = 0.00266), but by a slowing of opening rate (prolongation of $T_{ib}$) for T582I (*P* = 0.000865). Interestingly, for D984V an increased closing rate was partially compensated for by an increased opening rate (Fig. 4*B*, *bottom*).

### Strong potentiator responses of all five mutants assign even I336K and T465I to Class III

We next used macroscopic inside-out patch-clamp recordings to investigate to what extent the gating defects of G126D, T582I and D984V channels can be overcome by potentiator drugs. Besides VX-770, the corrector VX-445 was also shown to act as a co-potentiator that synergistically stimulates channel gating (Laselva et al., 2021; Shaughnessy et al., 2021; Veit et al., 2021). Pre-phosphorylated channels gating in 2 mM ATP were first exposed to 10 nM VX-770 (Fig. 5*A*–*C*), a clinically relevant free drug concentration (Matthes et al., 2016) which was found quasi-saturating for both WT CFTR and numerous mutants (Csanády & Torocsik, 2019; Li et al., 2024). Once the currents had stabilized, 1 μM VX-445 was also added (Fig. 5*A*–*C*) [EC$_{50}$ for co-potentiation by VX-445 is 1–15 nM for WT CFTR and various mutants (Laselva et al., 2021; Shaughnessy et al., 2021;

Veit et al., 2021)]. Subsequently, to maximally stimulate the channels, 300 nM PKA was re-applied and finally ATP replaced by 50 µM P-dATP, all in the maintained presence of both potentiator drugs (Fig. 5A–C). All three mutants were robustly and synergistically stimulated by both VX-770 and VX-445 (Fig. 5F; note logarithmic ordinate): VX-770 alone increased currents by ∼3–5-fold (Fig. 5F, 1st panel), and addition of VX-445 produced ∼2-fold additional potentiation (Fig. 5F, 2nd panel). Thus, total current stimulation by the combined pre-

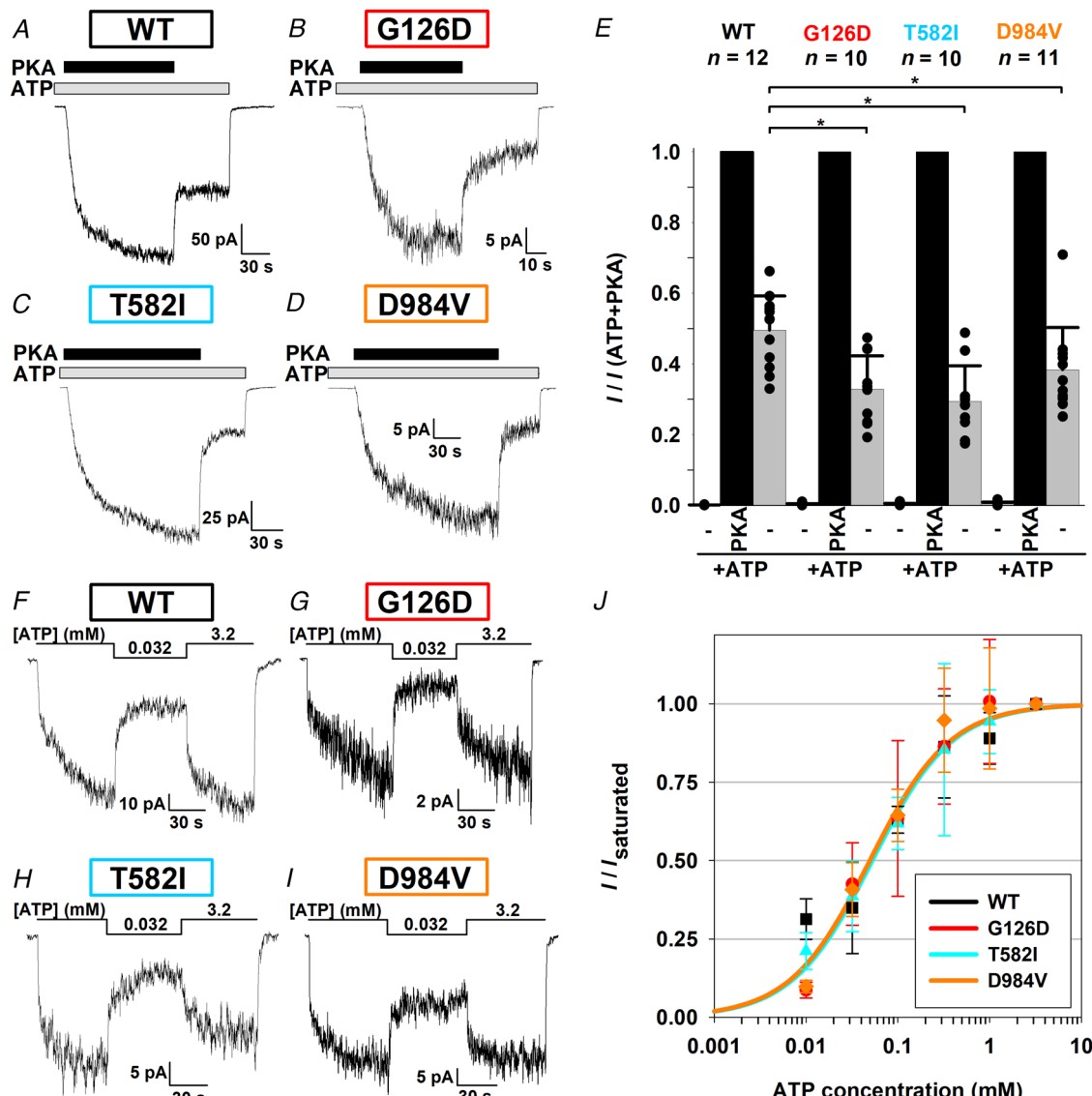

**Figure 3. PKA- and ATP-dependence of channel activity is intact for mutant CFTR constructs**
*A–D*, macroscopic inside-out patch currents for WT (*A*), G126D (*B*), T582I (*C*) and D984V (*D*) CFTR channels evoked by exposure to 300 nM PKA (*black bars*), in the presence of 2 mM ATP (*grey bars*). Membrane potential was −40 mV, and downward deflections correspond to channel openings. *E*, fractional channel activity before, during and after PKA exposure. Steady-state currents before PKA application, in the presence of PKA and following PKA removal are shown normalized to that in the presence of PKA (*black bars*). Bars represent mean ± SD with the number of patches indicated, and asterisks highlight statistically significant differences (*P* < 0.05). *F–I*, macroscopic current traces of pre-phosphorylated WT (*F*), G126D (*G*), T582I (*H*) and D984V (*I*) CFTR channels exposed to 32 µM ATP, bracketed by exposures to 3.2 mM ATP. Membrane potential was −40 mV, and downward deflections correspond to channel openings. The current segments shown were obtained following 1–3 min of exposure to 2 mM ATP + 300 nM PKA, as shown in *A–D*. *J*, ATP dose–response curves of the channel constructs, plotting currents in the presence of various test [ATP] normalized to the mean of the currents during bracketing exposures to 3.2 mM ATP. Symbols represent mean ± SD (*n* = 5–14). Coloured curves are fits to the Michaelis–Menten equation; $K_{1/2}$ was 50 ± 10, 53 ± 7, 53 ± 4 and 50 ± 7 µM, respectively, for WT, G126D, T582I and D984V CFTR.

sence of both drugs amounted to ~4-fold for D984V, ~6-fold for G126D and ~10-fold for T582I. However, open probability of all three mutants remained far from unity even in the presence of both potentiator drugs, as signalled by further marked current enhancements upon re-application of PKA and upon substitution of P-dATP for ATP (Fig. 5*F*, 3rd and 4th panels). Importantly, the observed maximal fractional stimulation of macroscopic current by the stimulator cocktail defines an upper bound for the initial open probability in just ATP. For T582I and D984V those upper bounds are consistent with the $P_o$ values estimated from single-channel recordings (Fig. 4*B*). However, for G126D the ~20-fold maximal stimulation (Fig. 5*F*, 4th panel, *red symbol*) implies that $P_o$ is $\leq$~0.05 under resting conditions, suggesting a slight overestimation of $P_o$ in the single-channel measurements (Fig. 4*B*), probably due to limitations in accurately determining the number of active channels in each patch.

For the I336K and T465I mutants our inability to elicit macroscopic currents by application of PKA + ATP was only partially explained by their observed maturation defects (Fig. 2*C*). We therefore assessed whether these mutants might also be impaired in gating, and to what extent they might respond to potentiator drugs. Indeed, exposure of pre-phosphorylated channels to 10 nM VX-770 + 1 μM VX-445 resulted in the activation of large macroscopic currents for both mutants (Fig. 5*D–E*), causing ~10-fold and ~80-fold current enhancement for I336K and T465I, respectively (Fig. 5*F*, 2nd panel). Moreover, both drug-stimulated currents were roughly doubled in the presence of PKA + P-dATP (Fig. 5*D–E*), resulting in maximal fractional stimulations of ~20-fold for I336K and ~200-fold for T465I CFTR. Thus, in the presence of just ATP, $P_o$ must be $\leq$~0.05 for phosphorylated I336K and $\leq$~0.005 for phosphorylated T465I CFTR channels, assigning both mutants also to Class III. Although for these two mutants even rough estimation of channel numbers is impossible, dwell-time analysis of segments of record in just ATP, prior to the addition of drugs (Fig. 5*D–E*, *initial segments*; cf. Fig. 6*A*), allows accurate estimation of closing rate. Combined with the above upper bounds for $P_o$, that information allows calculation of an upper bound for opening rate (see Methods). Such detailed kinetic analysis again revealed differential mechanisms underlying impaired gating for the two mutants. Whereas closing rate was accelerated ($T_b$ shortened) only modestly for T465I but robustly for I336K, opening rate was slowed ($T_{ib}$ prolonged) dramatically (>20-fold) for T465I but probably less so for I336K (Fig. 6*B*).

## Small reductions in unitary current amplitude assign G126D, I336K and T465I to Class IV

The single-channel current amplitudes for T582I and D984V CFTR were comparable to that of WT (Fig. 4*A*), but appeared visibly smaller for the other three mutants. Unitary conductances were therefore estimated by quantifying single-channel current amplitudes at −80, −40 and +40 mV membrane potentials (Fig. 7*A–D*), followed by fitting of unitary current–voltage relationships by linear regression to obtain slope conductances (Fig. 7*E*). Compared to WT CFTR (7.3 ± 0.2 pS), unitary conductances were indeed slightly but significantly reduced for G126D (6.1 ± 0.1 pS; $P = 0.00270$), I336K (5.0 ± 0.1 pS; $P = 0.00001$) and T465I (6.4 ± 0.2 pS; $P = 0.00976$), assigning these three mutants also to Class IV.

## Discussion

We have provided a detailed characterization of five rare CFTR mutations discovered in Hungarian CF patients, four of which had also been reported in other populations across Europe and Asia. For T582I, I336K and G126D, maturation defects were recently reported

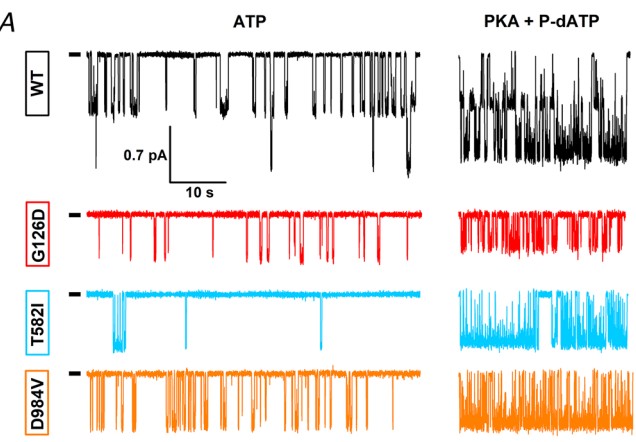

*A*

| Construct | n | Parameter | | |
|---|---|---|---|---|
| | | Opening rate (s⁻¹) | Closing rate (s⁻¹) | $P_o$ |
| **WT** | 11 | 0.45 ± 0.24 | 2.56 ± 1.53 | 0.16 ± 0.06 |
| **G126D** | 11 | 0.46 ± 0.36 | 5.11 ± 1.93 | 0.08 ± 0.05 |
| **T582I** | 7 | 0.07 ± 0.03 | 1.41 ± 0.54 | 0.04 ± 0.03 |
| **D984V** | 10 | 0.62 ± 0.37 | 4.22 ± 1.24 | 0.12 ± 0.07 |

**Figure 4. Single-channel recordings reveal reduced open probability for G126D and T582I CFTR**

*A*, representative current traces of two pre-phosphorylated WT, and single pre-phosphorylated G126D, T582I and D984V CFTR channels gating in the presence of 2 mM ATP (*left*), or during subsequent exposure to 50 μM P-dATP + 300 nM PKA (*right*). Membrane potential was −80 mV, downward deflections correspond to channel openings, and zero-current levels are indicated by short black dashes. The current segments shown were obtained following 1–3 min of exposure to 2 mM ATP + 300 nM PKA (cf. Fig. 3*A–D*). *B*, kinetic parameters of channel gating, obtained from single-channel measurements. Opening rate ($1/T_{ib}$; s⁻¹), closing rate ($1/T_b$; s⁻¹) and $P_o$: open probability. All values are shown as mean ± SD, and *n* represents the number of patches.

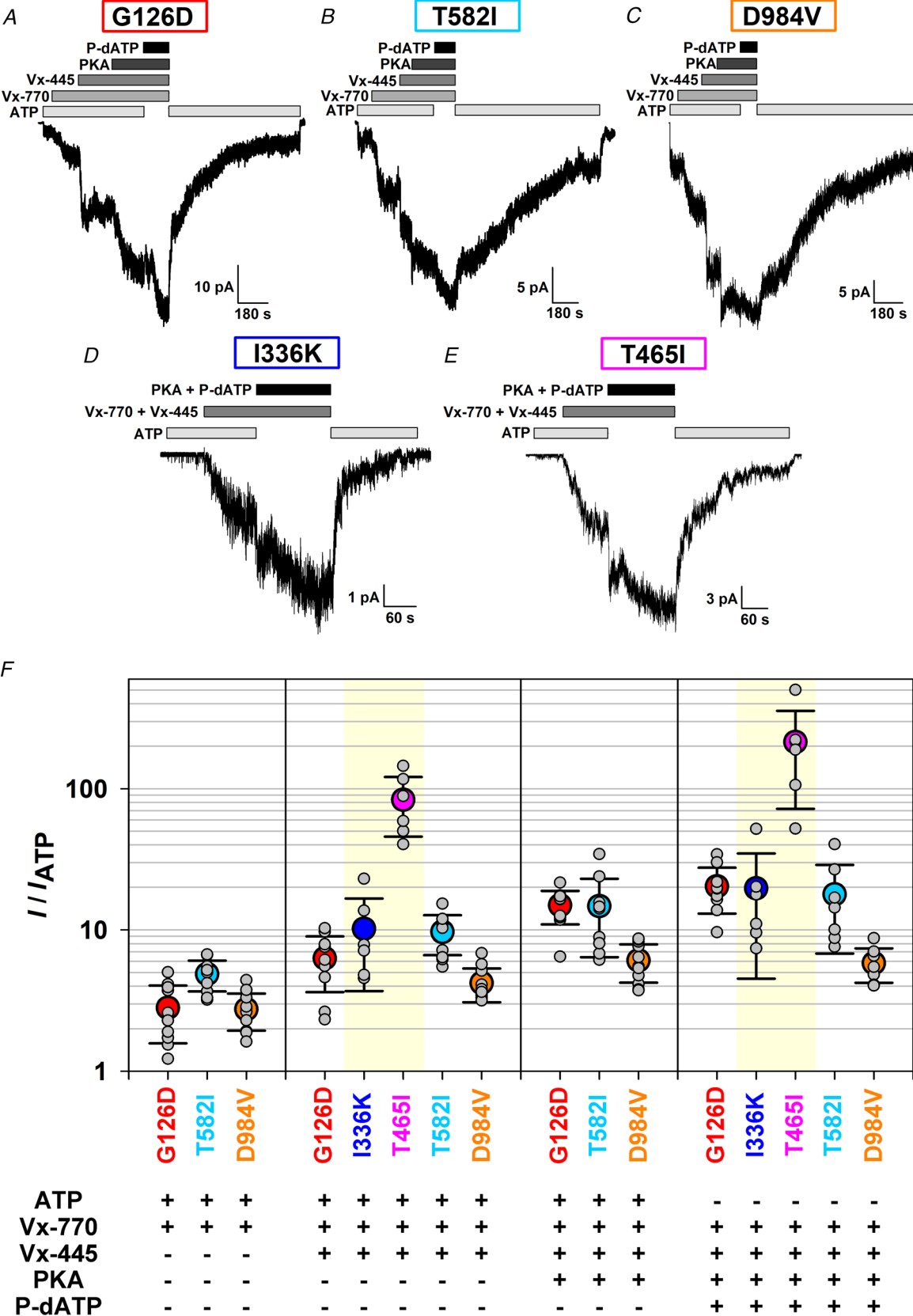

**Figure 5. All mutant channels respond robustly to CFTR potentiator drugs**

*A–E*, macroscopic currents of pre-phosphorylated G126D (*A*), T582I (*B*), D984V (*C*), I336K (*D*) and T465I (*E*) CFTR channels in 2 mM ATP, upon incremental exposure to an increasing number of the CFTR stimulators VX-770

(10 nM), VX-445 (1 μM), PKA (300 nM) and P-dATP (50 μM, replacing ATP). In *D–E*, VX-770 + VX-445 and PKA + P-dATP were introduced pairwise. Membrane potential was −40 mV, and downward deflections correspond to channel openings. The current segments shown were obtained following 1–3 min of exposure to 2 mM ATP + 300 nM PKA (cf. Fig. 3*A–D*). *F*, fractional steady-state currents of mutant CFTR channels upon incremental addition of the four potentiator compounds, normalized to that in the presence of ATP alone. Note logarithmic ordinate. Symbols plot mean ± SD (*n* = 7–11). *Faint yellow background* in panels 2 and 4 indicate a different experimental protocol (pairwise addition of stimulators) for mutants I336K and T465I.

(Bihler et al., 2024; Han et al., 2018; Hatton et al., 2022). I336K was found responsive to the corrector–potentiator combination ivacaftor + lumacaftor (Han et al., 2018), and both I336K and G126D were found responsive to ETI treatment in a screening study (Bihler et al., 2024), but no detailed functional information was available. The remaining two mutants, T465I and D984V, had never been functionally studied. Here we found that all five mutants form functional channels but exhibit complex phenotypes, assigning each of them into multiple mechanistic classes.

For all five variants, maturation was impaired to various degrees, as reported by defective glycosylation; the most severe disruption was observed for T465I (Fig. 2*A*, Band C; Fig. 2*C*, light *grey bars*). For T582I, I336K and G126D, the reductions in Band C intensities observed here were somewhat milder than those reported in the recent studies (Bihler et al., 2024; Han et al., 2018; Hatton et al., 2022). That discrepancy is probably due to the use of different

cell lines [HEK293-T cells here, *versus* HeLa, Fischer Rat Thyroid (FRT) and CFBE cells in the earlier studies], and the inherently semi-quantitative nature of densitometric analysis. Nevertheless, the conclusions of the four studies are convergent, assigning all three mutations to Class II. Importantly, for all five variants glycosylation could be restored to near-WT levels by the corrector drugs VX-661 + VX-445 (Fig. 2*C*, *dark grey bars*). As a cautionary note, since only crude membrane preparations were used here, our western blots confirm only improved maturation, but not actual translocation of the channels into the plasma membrane.

Activation by PKA remained intact, and both the reversible and irreversible component of channel activation could be clearly detected for all mutants (cf., Figs 3*A–D* and 5*D–E*). Interestingly, the fractional contribution of the irreversible component seemed significantly smaller for the mutants compared to WT (Fig. 3*E*, *grey bars*), suggesting that the mutations impair reversible stimulation (by PKA binding) less than irreversible stimulation (by phosphorylation). None of the mutations affected the apparent affinity ($K_{1/2}$) for ATP [cf. Fig. 3*F–J*; for I336K and T465I 2 mM ATP was also saturating: $I_{10 \text{ mM ATP}}/I_{2\text{mM ATP}}$ was 0.96 ± 0.09 (*n* = 3) and 0.90 ± 0.13 (*n* = 2), respectively]. Insofar as that $K_{1/2}$ reflects the ATP affinity of Site 2 in a closed channel (Vergani et al., 2003; Zhou et al., 2006), these data report no impairment by the mutations of ATP binding at Site 2.

All five mutations affected steady-state gating of phosphorylated channels, with T465I causing the largest decline in $P_o$, but the underlying changes in gating kinetics were clearly variable. The observed reduction in $P_o$ was caused primarily by an increased closing rate for G126D and I336K (Figs 4*B* and 6*B*), by a reduced opening rate for T582I (Fig. 4*B*), and by both effects for T465I (Fig. 6*B*). That pattern of kinetic changes is in agreement with the earlier finding that perturbations at the Site-2 NBD interface predominantly affect opening rate, those at the Site-1 NBD interface both opening and closing rate, whereas perturbations at the extracellular end of the TMDs primarily affect closing rate (Sorum et al., 2015, 2017). Interestingly, for D984V an accelerated closing rate was partially compensated for by an accelerated opening rate (Fig. 4*B*, *bottom*), suggesting a more complex effect of the latter mutation on channel gating energetics. A clear energetic interpretation of the above observations

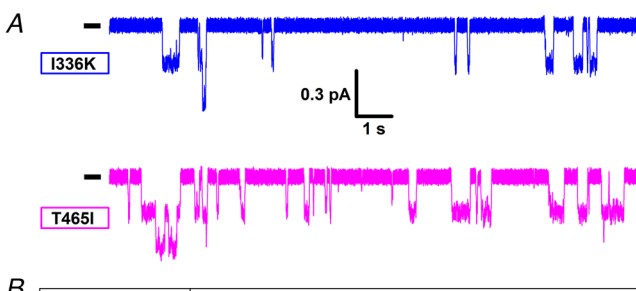

**A**

I336K

0.3 pA | 1 s

T465I

**B**

| Construct | Parameter | | |
|---|---|---|---|
| | Opening rate (s⁻¹) | Closing rate (s⁻¹) | $P_o$ |
| WT | 0.45 ± 0.24 | 2.56 ± 1.53 | 0.16 ± 0.07 |
| I336K | ≤ 0.2699 | 5.86 ± 3.63 | ≤ 0.0440 |
| T465I | ≤ 0.0161 | 3.10 ± 1.41 | ≤ 0.0052 |

**Figure 6. Single-channel gating parameters for mutants I336K and T465I**

*A*, resolvable unitary openings of pre-phosphorylated I336K and T465I CFTR channels gating in 2 mM ATP, from patches containing hundreds of channels, but prior to the addition of potentiator drugs (cf. Fig. 5*D–E*). Membrane potential was −40 mV, and downward deflections correspond to channel openings. The current segments shown were obtained following 1–3 min of exposure to 2 mM ATP + 300 nM PKA (cf. Fig. 3*A–D*). *B*, kinetic parameters of channel gating in 2 mM ATP for I336K and T465I CFTR. Opening rate (1/$T_{ib}$; s⁻¹), closing rate (1/$T_b$; s⁻¹) and $P_o$: open probability. Closing rate could be accurately estimated, and is shown as mean ± SD (*n* = 8 and 9 for I336K and T465I, respectively). An upper bound for $P_o$ was obtained from the maximal fractional stimulation by drugs (cf. Fig. 5*F*, *4th panel*), and an upper bound for opening rate was then calculated as described in the Methods. Gating parameters for WT are replotted from Fig. 4*B*.

would require detailed thermodynamic analysis based on investigation of multiple mutations at each target position, all in a non-hydrolytic background construct which gates at equilibrium (Sorum et al., 2015, 2017). However, such experiments are beyond the scope of the present study.

The multiplicative effects of VX-770, VX-445, PKA and P-dATP on macroscopic currents (Fig. 5) are consistent with distinct binding sites for the two drugs (Fiedorczuk & Chen, 2022), for PKA (Fiedorczuk et al., 2024) and for the nucleotide analogue (Fiedorczuk et al., 2024). Importantly, for all five mutants currents are robustly enhanced by VX-770 + VX-445, and reversible stimulation by PKA remains clearly observable even in the presence of potentiator drugs.

Mutations G126D, I336K and T465I caused a modest reduction in unitary conductance (Fig. 7). Unlike for G126D and I336K, which affect the pore region of the channel (Fig. 1*A*), for mutation T465I that finding was unexpected, as it targets a position in NBD1, ∼80 Å away from the selectivity filter. One possibility is that the mutation T465I alters intraburst gating, i.e. the kinetics of fast flickery closures. At the low bandwidth of our recordings an increased frequency, or duration, of flickery closures might appear as a slight reduction in unitary current amplitude. Of note, effects of mutations in Site 1 on intraburst gating kinetics have been documented (Chen et al., 2017), and a similar effect

was observed for WT channels gating in the presence of $N^6$-(2-phenylethyl)-ATP (P-ATP) (Csanády et al., 2013).

Although for all five variants reductions in $N$, $P_o$ or $i$ were observed, the severity of these effects was highly variable (Fig. 8). Grading each affected parameter as mildly (60–90% of WT; Fig. 8: ↓), substantially (30–60% of WT; Fig. 8: ↓↓) or severely (0–30% of WT; Fig. 8: ↓↓↓) reduced predicts the following rank order of the mutation-induced impairments in whole-cell $I_{CFTR}$: T465I ($N$↓↓↓, $P_o$↓↓↓, $i$↓) > I336K ($N$↓↓, $P_o$↓↓↓, $i$↓) > T582I ($N$↓, $P_o$↓↓↓) > G126D ($N$↓, $P_o$↓↓, $i$↓) > D984V ($N$↓↓, $P_o$↓). One limitation of these predictions is that the above biophysical parameters were quantified in heterologous expression systems, whereas patient-specific genetic and environmental factors may impact CFTR expression and function *in vivo* (Cutting, 2015). A further limitation of the predicted rank order is that these rare mutations are essentially never found in homozygous form. Thus, the severity of disease symptoms will also depend on the functional consequences of the mutation present on the other allele. The above rank order is expected only in cases where the above mutations are found *in trans* with a null allele, but published reports on clinical presentations for such patients are scarce. In one study I336K *in trans* with ΔF508 was found associated with both mild (pancreatic sufficient) and severe (pancreatic insufficient) forms of CF (Férec et al., 1993).

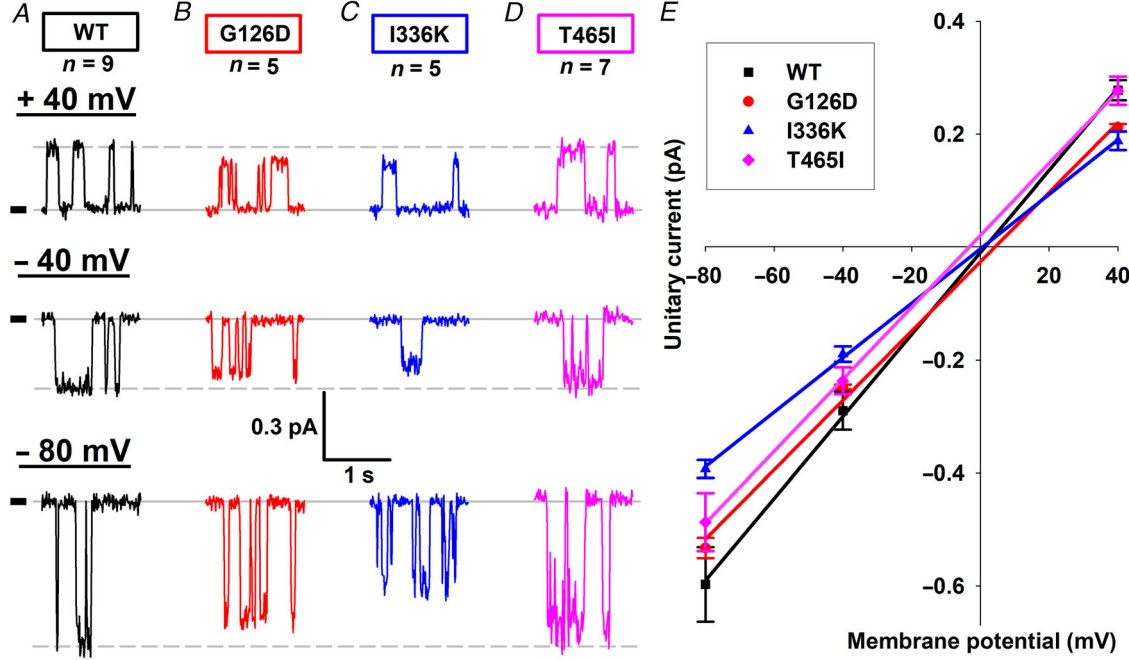

**Figure 7. Unitary conductance is reduced for G126D, I336K and T465I CFTR**
*A–D*, unitary channel openings recorded at +40, −40 and −80 mV membrane potential for WT (*A*), G126D (*B*), I336K (*C*) and T465I (*D*) CFTR; *n* represents the number of patches. *Short black dashes* and *continuous grey lines* indicate zero-current levels; *dashed grey lines* mark the open-channel current levels of wild-type CFTR. *E*, unitary current–voltage relationships (*symbols*) and straight-line fits (*lines*) yielding slope conductances of 7.2 ± 0.1, 6.2 ± 0.3, 5.0 ± 0.1 and 6.2 ± 0.1 pS for WT, G126D, I336K and T465I CFTR, respectively. Symbols plot mean ± SD.

| Mutation | Pathomechanism | | | Response to modulators | |
|---|---|---|---|---|---|
| | Class II (expression) | Class III (gating) | Class IV (permeation) | Correctors (Vx-445, Vx-661) | Potentiators (Vx-445, Vx-770) |
| G126D | ↓ | ↓↓ * | ↓ | ✓ | ✓✓ |
| I336K | ↓↓ | ↓↓↓ | ↓ | ✓✓ | ✓✓ |
| T465I | ↓↓↓ | ↓↓↓ | ↓ | ✓✓ | ✓✓✓ |
| T582I | ↓ | ↓↓↓ | → | ✓ | ✓✓ |
| D984V | ↓↓ | ↓ | → | ✓✓ | ✓ |

**Figure 8. Classification of the mutants based on pathomechanism and responses to modulators**
*Left*, summary of effects of the five mutations on channel expression/maturation (quantified by Band C density), gating (quantified by $P_o$) and permeation (quantified by unitary conductance). Each affected parameter was graded as mildly (60–90% of WT; ↓), substantially (30–60% of WT; ↓↓) or severely (0–30% of WT; ↓↓↓) impaired. *Right*, summary of responses of the five mutants to modulators. Enhancement of expression/maturation by correctors was categorized as not significant (✓) or significant (✓✓). Gating stimulation by potentiators was graded as modest (<5-fold; ✓), strong (5–20-fold; ✓✓) or excessive (>20-fold; ✓✓✓). *For G126D the $P_o = 0.08$ (Fig. 4*B*) is probably overestimated, given its ~20-fold stimulation by drugs (Fig. 5*F*, *4th panel*).

Importantly, all five variants proved responsive to the clinically used CFTR modulator drugs. The corrector combination VX-661 + Vx-445 improved glycosylation (Fig. 8: ✓), and this effect was significant (Fig. 8: ✓✓) for the three mutants that showed the most severe maturation defects (T465I, I336K and D984V). In addition, the potentiator combination VX-770 + VX-445 significantly stimulated the currents for all five variants. This effect can be graded as modest (<5-fold; Fig. 8: ✓) for D984V, strong (5–20-fold; Fig. 8: ✓✓) for G126D, I336K and T582I, or excessive (>20-fold; Fig. 8: ✓✓✓) for T465I. Thus, the data predict that the combined effect of ETI treatment should result in substantial fractional restoration of CFTR-mediated anion transport in CF patients carrying an allele with any of the five mutations.

In conclusion, our work justifies the recent endorsement of mutations G126D and I336K as ETI-eligible (https://pi.vrtx.com/files/uspi_elexacaftor_tezacaftor_ivacaftor.pdf). In addition, the evidence provided here suggests that ETI treatment is likely to be beneficial for patients carrying a T465I, T582I or D984V allele.

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

## Additional information

### Data availability statement

All data that support the findings of this study are available within the article.

## Competing interests

The authors declare no conflicts of interest.

## Author contributions

O.Z. and L.C. designed the research; O.Z. performed the research; O.Z. and L.C. analysed the data; O.Z. and L.C. wrote the paper. All authors have approved the final version of the manuscript and agreed to be accountable for all aspects of the work. All persons designated as authors qualify for authorship, and all those who qualify for authorship are listed.

## Funding

This study was supported by the EU Horizon 2020 Research and Innovation Program grant 739593, Cystic Fibrosis Foundation Research Grant CSANAD21G0, and National Research, Development and Innovation Fund grant KKP 144199 to L.C.

## Acknowledgements

We thank Drs Martina Gentzsch and Tim Jensen for providing CFTR antibodies, and Dr Csaba Mihályi for help with data export from densitometric images.

## Keywords

CFTR, cystic fibrosis, elexacaftor–tezacaftor–ivacaftor (ETI, Trikafta, Kaftrio), rare mutation

## Supporting information

Additional supporting information can be found online in the Supporting Information section at the end of the HTML view of the article. Supporting information files available:

**Peer Review History**

