## [Peer Review History · The Journal of Physiology]

Molecular and pharmacological evaluation of rare, cystic fibrosis causing missense mutations of the CFTR channel

Olivér Závoti and László Csanády

DOI: 10.1113/JP288955

Corresponding author(s): László Csanády (csanady.laszlo@semmelweis.hu)

Review Timeline:

Submission Date:	25-Mar-2025
Editorial Decision:	25-Apr-2025
Revision Received:	15-May-2025
Editorial Decision:	04-Jun-2025
Revision Received:	04-Jun-2025
Accepted:	02-Jul-2025

Senior Editor: *Peying Fong*

Reviewing Editor: *Tzyh-Chang Hwang*

Transaction Report:

Dear Dr Csanády,

Re: JP-RP-2025-288955 "Molecular and pharmacological evaluation of rare, cystic fibrosis causing missense mutations of the CFTR channel" by Olivér Závoti and László Csanády

Thank you for submitting your manuscript to The Journal of Physiology. It has been assessed by a Reviewing Editor and by 2 expert referees and we are pleased to tell you that it is acceptable for publication following satisfactory revision.

REVISION CHECKLIST:

We look forward to receiving your revised submission.

Yours sincerely,

Peying Fong
Senior Editor
The Journal of Physiology

REQUIRED ITEMS

- Author photo and profile. First or joint first authors are asked to provide a short biography (no more than 100 words for one author or 150 words in total for joint first authors) and a portrait photograph. These should be uploaded and clearly labelled together in a Word document with the revised version of the manuscript. See Information for Authors for further details.

- You must start the Methods section with a paragraph headed Ethical Approval. If experiments were conducted on humans, confirmation that informed consent was obtained, preferably in writing, that the studies conformed to the standards set by the latest revision of the Declaration of Helsinki and that the procedures were approved by a properly constituted ethics committee, which should be named, must be included in the article file. If the research study was registered (clause 35 of the Declaration of Helsinki), the registration database should be indicated, otherwise the lack of registration should be noted as an exception (e.g. The study conformed to the standards set by the Declaration of Helsinki, except for registration in a database). For further information see: <https://physoc.onlinelibrary.wiley.com/hub/human-experiments>.

- Your manuscript must include a complete Additional Information section, including competing interests; funding; author contributions and acknowledgements.

- Please upload separate high-quality figure files via the submission form.

- Please ensure that any tables are editable and in Word format, and wherever possible, embedded in the article file itself.

- Please ensure that the Article File you upload is a Word file.

- Papers must comply with the Statistics Policy: https://jp.msubmit.net/cgi-bin/main.plex?form_type=display_requirements#statistics.

In summary:

- If $n \leq 30$, all data points must be plotted in the figure in a way that reveals their range and distribution. A bar graph with data points overlaid, a box and whisker plot or a violin plot (preferably with data points included) are acceptable formats.

- If $n > 30$, then the entire raw dataset must be made available either as supporting information, or hosted on a not-for-profit repository, e.g. FigShare, with access details provided in the manuscript.

- 'n' clearly defined (e.g. x cells from y slices in z animals) in the Methods. Authors should be mindful of pseudoreplication.

- All relevant 'n' values must be clearly stated in the main text, figures and tables.

- The most appropriate summary statistic (e.g. mean or median and standard deviation) must be used. Standard Error of the Mean (SEM) alone is not permitted.

- Exact p values must be stated. Authors must not use 'greater than' or 'less than'. Exact p values must be stated to three significant figures even when 'no statistical significance' is claimed.

- Please include an Abstract Figure file, as well as the Figure Legend text within the main article file. The Abstract Figure is a piece of artwork designed to give readers an immediate understanding of the research and should summarise the main conclusions. If possible, the image should be easily 'readable' from left to right or top to bottom. It should show the physiological relevance of the manuscript so readers can assess the importance and content of its findings. Abstract Figures should not merely recapitulate other figures in the manuscript. Please try to keep the diagram as simple as possible and without superfluous information that may distract from the main conclusion(s). Abstract Figures must be provided by authors no later than the revised manuscript stage and should be uploaded as a separate file during online submission labelled as File Type 'Abstract Figure'. Please also ensure that you include the figure legend in the main article file. All Abstract Figures should be created using BioRender. Authors should use The Journal's premium BioRender account to export high-resolution images. Details on how to use and access the premium account are included as part of this email.

EDITOR COMMENTS

Reviewing Editor:

This manuscript provided biochemical and electrophysiological data supporting the usefulness of ETI in the treatment of patients carrying five rare CF-causing mutations found in Hungarian population. The results are convincing and potentially important in clinical application.

In addition to the questionable structural interpretations of the residues investigated in the current study as pointed out by the second reviewer, I have a few concerns over the electrophysiological data. First, as pointed out by the second reviewer, VX-770 did not show any effect on G126D in Figure 5A contrary to Figure 5F. Second, I wonder why the protocol in Figure 5 varies. Even more puzzling is the combination of PKA and P-dATP as the lead author's lab showed previously that P-ATP cannot be used by PKA. If the attempt is not about phosphorylation, then they should be applied separated as in Figure 5A, B, and C. Third, as pointed out by the second reviewer, Figure 5F needs some clarification. If VX-770 and VX-445 were applied together as shown in Figure 5D and E, whereas they were applied separately in Figure 5A, B, and C, it seems inappropriate to put all the data in the same category in Figure 5F. Same issue applies to PKA + P-dATP. Third, I wonder if placing T465I in Class IV mutation is appropriate (Figure 7) because the effect of the mutation on single-channel amplitude may well lie in the variation of experiments. Moreover, the residue resides in NBD1; it seems hard to explain this effect if any.

Senior Editor:

The initial review of your manuscript, "Molecular and pharmacological evaluation of rare, cystic fibrosis causing missense mutations of the CFTR channel", is now complete. As evidenced by the collected, detailed reports from two Expert Referees, as well as the Reviewing Editor, the topic has generated substantial interest. I view this as indicative of your work's potential to make a significant impact and encourage you to review their comments carefully. Many are directed at improving clarity of readability and interpretation of the findings. While you will see that the concerns raised by Referee 2 regarding structural interpretation are of utmost priority, you are encouraged to address all comments raised by both Referees, as well as those of the Reviewing Editor, as you prepare your revised manuscript.

Please also note the need for further details regarding the methods for both anaesthesia and handling, as well as euthanasia, that currently are simply referred to as being compliant with procedures given in their Institutional Care and Use of Animals protocol.

Regarding the statistics, journal policy is to use SD so please change this where you are currently reporting SEM.

REFEREE COMMENTS

Referee #1:

CFTR modulators approved for clinical use are transforming the lives of most people with cystic fibrosis (CF). However, people with CF and many rare CF-causing CFTR mutations are currently ineligible for these CFTR modulators because

their response to these drugs is unknown. In the present study, the authors investigate five rare CF-causing CFTR mutations found in the Hungarian population: G126D, I336K, T465I, T582I and D984V. For their studies, the authors investigate CFTR protein maturation in HEK 293T cells transiently expressing the mutations and channel function in excised inside-out membrane patches from *Xenopus* oocytes transiently expressing the mutations. When compared with wild-type CFTR, I336K and particularly T465I decreased the amount of mature CFTR protein produced, with the other mutations causing less marked reductions. However, the CFTR correctors elexacaftor and tezacaftor restored wild-type or near wild-type levels of mature CFTR protein to all the mutations. Although the five mutations formed Cl⁻ channels regulated by PKA-dependent phosphorylation and intracellular ATP, analysis of channel gating revealed complex effects of the mutations on the opening and closing rates with I336K and T465I causing the greatest reduction in open probability. Nevertheless, channel activity was substantially restored to the mutations using the CFTR potentiators ivacaftor and elexacaftor. Finally, the authors reveal that the mutations G126D, I336K and T465I caused some reduction in the single-channel conductance of CFTR. By grading both the loss and recovery of CFTR expression and function, the authors classify the impact of mutations and CFTR modulators on CFTR expression and function. Their data reveal that all mutations have multiple mechanisms of CFTR dysfunction, but that clinically-approved CFTR modulators benefit all the mutations.

The authors' experiments are well-designed, the data are clear and their interpretation logical.

General comments

A valuable addition to this manuscript would be information about the clinical phenotype of people with CF and the five rare CF-causing missense mutations studied. Are these mutations associated with classical or non-classical CF? If this information is available, the Discussion should be revised to discuss whether the effects of the mutations on CFTR expression and function explain the clinical phenotype.

The authors introduce the five rare CF-causing missense mutations using their official cDNA and protein names, but subsequently use legacy names to refer to the mutations. Please indicate the legacy names of the mutations when they are first introduced.

Please revise the Discussion to discuss the potential limitations of the present study.

Please revise Figure 2C and D, Figure 3E, Figure 5F to include keys to identify the different shaded bars and coloured symbols. Thank you.

Please revise Figure 2C and D, Figure 3E and J, Figure 5F and Figure 6E to show error bars for SD only. The inclusion of additional bars to show SEM obscures the data. Thank you.

Please check the references for author spelling errors and missing information. Thank you.

Specific comments

P2, abstract: The abstract lacks balance. About two-thirds of the abstract is occupied by Background and Methods. The description of Results is incomplete with no reference to the effects of mutations on anion permeation. Please revise the abstract to provide a balanced summary of the work.

P2, Keywords: For ETI, please give the name of all three drugs and make clear that Trikafta and Kaftrio refer to ETI and are not different keywords.

P3, Introduction, 11-6: The first two sentences lack supporting references. The third sentence refers to the channel function of

CFTR, but the reference precedes the identification of CFTR as an anion channel. Please provide suitable references for these sentences.

P3, Introduction, I10: "ABC signature sequence".

P3, Introduction, I25-26: "CF affects multiple organs ...".

P3, Introduction, I28: Rather than cite a review, please cite the original references for the six classes of CFTR mutations (Welsh MJ, Smith AE. *Cell*. 1993 Jul 2;73(7):1251-4; Zielenski J, Tsui LC. *Annu Rev Genet*. 1995;29:777-807; Haardt M et al. *J Biol Chem*. 1999 Jul 30;274(31):21873-7).

P4, Introduction, I6: F508del also disrupts the stability of CFTR protein at the plasma membrane, leading to its classification as a class II/III/VI mutation (Lukacs GL et al. *J Biol Chem*. 1993 Oct 15;268(29):21592-8).

P4, Introduction, I9: Po was defined on page 3, I21.

P5, Methods, I1 and P6, Methods, I28: "CFTR/pcDNA3".

P6, Methods, I18 and I22: "boundaries" and "boundary".

P7, Methods, I23: Please indicate that the anti-CFTR monoclonal antibody 5E2 recognises the C-terminal portion of NBD2 (residues 1371-1385).

P10, Results, I1-2: Please refer to the locations of CF-causing mutations rather than target positions.

P10, Results, I2-3 and I19-20: Please indicate that PDB id: 6MSM and PDB id: 5UAK are human CFTR.

P10, Results, I19-20: "dephosphorylated ATP-free, "closed" human CFTR".

P11, Results, I16-25: Please refer the Reader to Figure 2B and 2D when describing the effects of elexacaftor and tezacaftor on the maturation of the mutants studied.

P11, Results, I28: Figure 3A-D shows data for three of the five mutants studied. To avoid potential confusion, please delete "(Fig. 3A-D)".

P13, Results, I3-5 and Figure 4B: The authors indicate that the increased closing rate of D984V is partially compensated by an increased opening rate. Although not as marked, the opposite appears to be the case for T582I, where a slower opening rate is offset to some extent by a slower closing rate.

P13, Results, I10-15: Please refer the Reader to Figure 5A-C in both these sentences.

P13, Results, I26: "boundaries".

P14, Results, I15 and I16: "boundaries" and "boundary".

P14, Results, I21-27: Please revise the beginning of this section to indicate that T582I and D984V were without effect on single-channel current amplitude, but G126D, I336K and T465I reduced single-channel current amplitude and refer the Reader to Figures 4 and S1.

P15, Discussion, I3-5: I336K was studied by Han ST et al. JCI Insight. 2018 Jul 26;3(14):e121159. These authors showed that it had a severe maturation defect when expressed in CF bronchial epithelial cells with CFTR function increased around 10 fold with lumacaftor and ivacaftor in Ussing chamber studies. These authors also showed that residue I336 faced away from the channel pore.

P15, Discussion, I20-21: Please refer the Reader to Figure 5D and E for PKA activation of I336K and T465I.

P17, Discussion, I16-17: Please provide a reference for people with CF and G126D and I336K now being eligible for treatment with ETI. A web address will suffice.

P18, Figure legends, I2-3: Please indicate that PDB id: 6MSM is human CFTR.

P18, Figure legends, I5-6: Please refer to panels B1 - B6.

P20, Figure legends, I11-12: For the reference to ~20-fold stimulation by drugs, don't you mean Fig. 5F, 4th panel?

P20, Figure legends, I19: "Fig. 5F, 4th panel" and "boundary".

Figure 3A-D and F-I: Please revise the Figure Legend to indicate that downward deflections correspond to channel openings.

Figure 4A: (i) The inclusion of current amplitude scale bars for wild-type and each of the three mutants is potentially confusing. It would be preferable to have one scale bar for current amplitude as for the single scale bar for time (see also Fig. S1A). Is the G126D recording drawn to the same scale as wild-type, T582I and D984V? (ii) Please indicate the closed channel level.

Figure 5A-E: Please revise the Figure Legend to indicate that downward deflections correspond to channel openings.

Figure 6A-D: Please revise the Figure Legend to indicate that the continuous and dashed grey lines indicate the closed and

open channel levels of wild-type CFTR.

Figure 6E: Please revise the straight line fits so that they do not extend beyond +40 mV.

Referee #2:

See attached file for comments.

END OF COMMENTS

In the manuscript titled 'Molecular and pharmacological evaluation of rare, cystic fibrosis causing missense mutations of the CFTR channel', the authors sort to characterize the molecular and pharmacological effects of 5 rare CF-causing mutations. They used western blot to show that all five mutations decrease the quantity of mature CFTR protein which can be corrected by CFTR corrector VX-445 and VX-661. Micro- and macroscopic recordings are shown to demonstrate their gating and conductance defects. Potentiators VX-770 and VX-445 are shown to rescue the gating defects found which provides 'ADDITIONAL' evidence that ETI treatment can be used for patients carrying these CF-causing mutations. ETI has been approved as a treatment for patients carrying G126D and I336K by US-FDA about one year ago, which means Vertex Pharmaceuticals and US-FDA already had the data that point to its effectiveness on these mutants. This reduces the significance/novelty of the data shown here. However, given that those data are not publicly available, current data can still benefit the CF/CFTR community, especially there are three other mutants involved here. Here are the comments that can help the authors to revise their manuscript:

1. As abovementioned earlier approval, the author may want to revise what they mention in the second paragraph on page 4 that goes 'However, for many rare variants, ETI ...' to make their study more justified, although they seem to have realized this in the last paragraph of the manuscript.
2. In the second paragraph of 'Results', the authors claimed that T582 side chain forms a H-bond with the side chain of D579 and cited Zhang et al., 2018. However, upon examination of the Cryo-EM map and model from that publication, side chain information is missing for D579 in the Cryo-EM map, which indicates the assignment of the side chain orientation is not accurate, that further means this datum does not really support such claim. Nonetheless, in another map published by Gao et al., 2024, the E1371Q-CFTR map clear shows side chain density of both residues, which readily support the author's claim. See below the two figures for comparison. The map published by Gao et al. is a better one in terms of supporting the author's claim here and hence the authors should cite this one here, instead of Zhang et al., 2018.

3. In the same paragraph, the authors claimed in the model 5UAK, 582-579 and 984-289 move further apart from each other and don't form the side interactions. Upon examination of the map and model, in that Cryo-EM map, densities for 582 and 579 are both missing, so no one really knows whether they form a salt bridge or not. Actually, the distance shows they may form a salt bridge, if the authors choose to believe the model. Similarly, in that map, while density for 984 side chain is bad, density for 289 side chain is well defined, meaning they still can possibly form a salt bridge. See the figures below. Therefore, the claims the authors made here are based on incomplete information and need to be revised to reflect rigor of interpretation of the data.

4. In figure 2C, the WT has two grey bars, is the right side one with or without correctors? If with correctors, change the color to dark grey, if not, then it is not meaningful to put it side by side there. This way gives readers the impression that correctors don't work on WT CFTR. Also why are both SD and SEM shown?
5. Figure 5A does not show VX-770 increase channel activity for G126D, but the authors claim so in the text. No steady state is seen in this recording too. They need to either revise their interpretation of their data or re-examine their data. In the same figure, I recommend the authors to change the abscissa of the table to better reflect each condition – e.g., use VX-770 + VX-445, instead of only VX-445.
6. As these data intend to provide evidence to justify prescription of ETI for patients with the mutations studied here, the reviewer is wondering why all potentiator experiments are done with pre-phosphorylated CFTR variants, which does not reflect real case where in vivo CFTR are usually phosphorylated and have ATP around. The authors need to repeat experiments in Figure 5 with phosphorylated CFTR, data from which will help to better serve the purposes of this study.
7. On page 16, G126D is at the periphery of the molecule, rather than in the pore region, how can this mutation affect the pore region of the channel?

8. On page 17, the whole cell current prediction seems too simplified, and the actual significance of this prediction is not clear. If the authors want to make these specific claims, they need to do whole cell experiments to verify they are true or not.
9. On page 12, is it Figure 4B, right, instead of figure 2B, right, in the bottom line? In the following paragraph, I suppose there is a typo: 'for' , not 'or' before T582I.

Reviewing Editor:

This manuscript provided biochemical and electrophysiological data supporting the usefulness of ETI in the treatment of patients carrying five rare CF-causing mutations found in Hungarian population. The results are convincing and potentially important in clinical application.

We thank the Reviewing Editor for the appreciative words.

In addition to the questionable structural interpretations of the residues investigated in the current study as pointed out by the second reviewer,

Amended, see responses to Reviewer 2.

I have a few concerns over the electrophysiological data. First, as pointed out by the second reviewer, VX-770 did not show any effect on G126D in Figure 5A contrary to Figure 5F.

We agree that the trace shown was not representative for the VX-770 response (it was the trace that provided the lowest gray symbol for the 1st panel of Fig. 5F). We have replaced the recording in Fig. 5A with a more representative sample trace.

Second, I wonder why the protocol in Figure 5 varies... If VX-770 and VX-445 were applied together as shown in Figure 5D and E, whereas they were applied separately in Figure 5A, B, and C, it seems inappropriate to put all the data in the same category in Figure 5F. Same issue applies to PKA + P-dATP.

For the two mutants I336K and T465I we could never see macroscopic currents in just ATP, following phosphorylation by PKA. For these two mutants we therefore employed a simplified protocol, introducing the stimulants pairwise, rather than one at a time. Fig. 5F shows, for all conditions, fractional stimulation normalized to the activity (mean current) in ATP alone, observed in the same patch. Thus, showing fractional stimulation by VX-770+VX-445 in the same panel for all five mutants, regardless of whether the drugs had been introduced simultaneously or following each other, is a fair comparison. The same applies for PKA and P-dATP (also see explanation below). Nevertheless, to emphasize the differences in protocol, in Fig. 5F we have now highlighted the data for I336K and T465I with a faint yellow background, and also draw the readers attention to the protocol difference in the figure legend.

Even more puzzling is the combination of PKA and P-dATP as the lead author's lab showed previously that P-ATP cannot be used by PKA. If the attempt is not about phosphorylation, then they should be applied separated as in Figure 5A, B, and C.

P-ATP can indeed not be used for phosphorylation by PKA. However, as explained in the Methods text (and now also specified in the legend for Fig. 5), prior to the experimental segments shown in Fig. 5A-E, the channels had already been phosphorylated by exposure to 2 mM ATP + 300 nM PKA in each case (cf., Fig. 3A-D).

PKA and P-dATP stimulate currents independently of each other. Re-application of PKA in Fig. 5A-E enhances currents of already phosphorylated channels merely through reversible stimulation, caused by PKA binding, which happens even in the presence of P-ATP (or P-dATP). On the other hand, P-dATP directly stimulates channel gating by binding to CFTR's NBDs (Miki et al., 2010, J. Biol. Chem., 285: 19967-19975). This is now explained in the text describing the single-channel experiments in Figure 4, where we used the same strategy to facilitate channel counting (P14, bottom): *"In an attempt to estimate the number of channels in the patch, at the end of each recording 300 nM PKA was reapplied to provide reversible stimulation, and ATP was replaced with 50 μ M 2'-deoxy-N⁶-(2-phenylethyl)-ATP (P-dATP) which stimulates CFTR gating by binding directly to CFTR's NBDs"*

(Miki et al., 2010). This combination results in substantial gating stimulation (Sorum et al., 2017) (Fig. 4A, right)."

For the aim in Fig. 5, to eventually boost channel activity maximally, it is indifferent whether PKA and P-dATP are applied simultaneously or sequentially.

Third, as pointed out by the second reviewer, Figure 5F needs some clarification.

We have added more extensive labeling to clarify the conditions.

Third, I wonder if placing T465I in Class IV mutation is appropriate (Figure 7) because the effect of the mutation on single-channel amplitude may well lie in the variation of experiments. Moreover, the residue resides in NBD1; it seems hard to explain this effect if any.

The impact of this mutation on unitary conductance was indeed small, but it proved to be statistically significant ($p=0.000976$), i.e., not within the variation of experiments. We mention it for the sake of completeness, even if we cannot provide an explanation for the observed phenomenon. (One potential explanation is mentioned in Discussion (P18): "*One possibility is that mutation T465I alters intraburst gating, i.e., the kinetics of fast flickery closures. At the low bandwidth of our recordings an increased frequency, or duration, of flickery closures might appear as a slight reduction in unitary current amplitude. Of note, effects of mutations in Site 1 on intraburst gating kinetics have been documented (Chen et al., 2017), and a similar effect was observed for WT channels gating in the presence of N6-(2-phenylethyl)-ATP (P-ATP) (Csanády et al., 2013).*")

Senior Editor:

The initial review of your manuscript, "Molecular and pharmacological evaluation of rare, cystic fibrosis causing missense mutations of the CFTR channel", is now complete. As evidenced by the collected, detailed reports from two Expert Referees, as well as the Reviewing Editor, the topic has generated substantial interest. I view this as indicative of your work's potential to make a significant impact and encourage you to review their comments carefully. Many are directed at improving clarity of readability and interpretation of the findings. While you will see that the concerns raised by Referee 2 regarding structural interpretation are of utmost priority, you are encouraged to address all comments raised by both Referees, as well as those of the Reviewing Editor, as you prepare your revised manuscript.

We thank the Senior Editor for the positive evaluation. We have addressed all the comments of the Referees and the Reviewing Editor, as explained in detail in our responses.

Please also note the need for further details regarding the methods for both anaesthesia and handling, as well as euthanasia, that currently are simply referred to as being compliant with procedures given in their Institutional Care and Use of Animals protocol. Regarding the statistics, journal policy is to use SD so please change this where you are currently reporting SEM.

Done, thank you. Furthermore, to conform to journal policy, we have moved Fig. S1 into the main text (new Fig. 6). Previous Figures 6 and 7 have been renumbered to 7 and 8, respectively.

REFEREE COMMENTS

Referee #1:

CFTR modulators approved for clinical use are transforming the lives of most people with cystic

fibrosis (CF). However, people with CF and many rare CF-causing CFTR mutations are currently ineligible for these CFTR modulators because their response to these drugs is unknown. In the present study, the authors investigate five rare CF-causing CFTR mutations found in the Hungarian population: G126D, I336K, T465I, T582I and D984V. For their studies, the authors investigate CFTR protein maturation in HEK 293T cells transiently expressing the mutations and channel function in excised inside-out membrane patches from *Xenopus* oocytes transiently expressing the mutations. When compared with wild-type CFTR, I336K and particularly T465I decreased the amount of mature CFTR protein produced, with the other mutations causing less marked reductions. However, the CFTR correctors elexacaftor and tezacaftor restored wild-type or near wild-type levels of mature CFTR protein to all the mutations. Although the five mutations formed Cl⁻ channels regulated by PKA-dependent phosphorylation and intracellular ATP, analysis of channel gating revealed complex effects of the mutations on the opening and closing rates with I336K and T465I causing the greatest reduction in open probability. Nevertheless, channel activity was substantially restored to the mutations using the CFTR potentiators ivacaftor and elexacaftor. Finally, the authors reveal that the mutations G126D, I336K and T465I caused some reduction in the single-channel conductance of CFTR. By grading both the loss and recovery of CFTR expression and function, the authors classify the impact of mutations and CFTR modulators on CFTR expression and function. Their data reveal that all mutations have multiple mechanisms of CFTR dysfunction, but that clinically-approved CFTR modulators benefit all the mutations.

The authors' experiments are well-designed, the data are clear and their interpretation logical.

We thank the Reviewer for the appreciative summary.

General comments

A valuable addition to this manuscript would be information about the clinical phenotype of people with CF and the five rare CF-causing missense mutations studied. Are these mutations associated with classical or non-classical CF? If this information is available, the Discussion should be revised to discuss whether the effects of the mutations on CFTR expression and function explain the clinical phenotype.

We have added a paragraph to discuss these issues (P19). A limitation for correlating the clinical phenotype with the *in vitro* data is that these rare mutations are essentially never found in homozygous form. Thus, the severity of disease symptoms will also depend on the functional consequences of the mutation present on the other allele. The rank order predicted by our *in vitro* data is expected to hold only for patients that carry the above mutations in trans with a null allele, but published reports on clinical presentations for such patients are scarce. In one study I336K in trans with Δ F508 was found associated with both mild (pancreatic sufficient) and severe (pancreatic insufficient) forms of CF (Férec et al., 1993).

The authors introduce the five rare CF-causing missense mutations using their official cDNA and protein names, but subsequently use legacy names to refer to the mutations. Please indicate the legacy names of the mutations when they are first introduced.

Done, thank you.

Please revise the Discussion to discuss the potential limitations of the present study.

Done, thank you (P19, 2nd par).

Please revise Figure 2C and D, Figure 3E, Figure 5F to include keys to identify the different shaded bars and coloured symbols. Thank you.

Done, thank you.

Please revise Figure 2C and D, Figure 3E and J, Figure 5F and Figure 6E to show error bars for SD only. The inclusion of additional bars to show SEM obscures the data. Thank you.

Done, thank you.

Please check the references for author spelling errors and missing information. Thank you.

Done, thank you.

Specific comments

P2, abstract: The abstract lacks balance. About two-thirds of the abstract is occupied by Background and Methods. The description of Results is incomplete with no reference to the effects of mutations on anion permeation. Please revise the abstract to provide a balanced summary of the work.

Done, thank you.

P2, Keywords: For ETI, please give the name of all three drugs and make clear that Trikafta and Kaftrio refer to ETI and are not different keywords.

Done, thank you.

P3, Introduction, l1-6: The first two sentences lack supporting references. The third sentence refers to the channel function of CFTR, but the reference precedes the identification of CFTR as an anion channel. Please provide suitable references for these sentences.

Done, thank you.

P3, Introduction, l10: "ABC signature sequence".

Done, thank you.

P3, Introduction, l25-26: "CF affects multiple organs ...".

Done, thank you.

P3, Introduction, l28: Rather than cite a review, please cite the original references for the six classes of CFTR mutations (Welsh MJ, Smith AE. Cell. 1993 Jul 2;73(7):1251-4; Zielenski J, Tsui LC. Annu Rev Genet. 1995;29:777-807; Haardt M et al. J Biol Chem. 1999 Jul 30;274(31):21873-7).

Done, thank you.

P4, Introduction, l6: F508del also disrupts the stability of CFTR protein at the plasma membrane, leading to its classification as a class II/III/VI mutation (Lukacs GL et al. J Biol Chem. 1993 Oct 15;268(29):21592-8).

We have added this information with the suggested reference.

P4, Introduction, l9: Po was defined on page 3, l21.

Corrected, thank you.

P5, Methods, I1 and P6, Methods, I28: "CFTR/pcDNA3".

Corrected, thank you.

P6, Methods, I18 and I22: "boundaries" and "boundary".

P13, Results, I26: "boundaries".

P14, Results, I15 and I16: "boundaries" and "boundary".

P20, Figure legends, I19: "boundary".

We have kept our original wording ("upper bound(s)") in all these instances, as we believe that to be the appropriate term here. An upper bound is a generally accepted mathematical term, meaning a number that is larger or equal than any member of a given set.

P7, Methods, I23: Please indicate that the anti-CFTR monoclonal antibody 5E2 recognises the C-terminal portion of NBD2 (residues 1371-1385).

Done, thank you.

P10, Results, I1-2: Please refer to the locations of CF-causing mutations rather than target positions.

Done, thank you.

P10, Results, I2-3 and I19-20: Please indicate that PDB id: 6MSM and PDB id: 5UAK are human CFTR.

Done, thank you.

P10, Results, I19-20: "dephosphorylated ATP-free, "closed" human CFTR".

Done, thank you.

P11, Results, I16-25: Please refer the Reader to Figure 2B and 2D when describing the effects of elxacaftor and tezacaftor on the maturation of the mutants studied.

Done, thank you.

P11, Results, I28: Figure 3A-D shows data for three of the five mutants studied. To avoid potential confusion, please delete "(Fig. 3A-D)".

Done, thank you.

P13, Results, I3-5 and Figure 4B: The authors indicate that the increased closing rate of D984V is partially compensated by an increased opening rate. Although not as marked, the opposite appears to be the case for T582I, where a slower opening rate is offset to some extent by a slower closing rate.

We agree. However, in contrast to D984V, for T582I the resulting P_o remains significantly lower than for WT ($p=0.000195$). In Discussion (P18) we do provide a more extensive comparison of the pattern of mutation-induced changes in gating kinetics.

P13, Results, I10-15: Please refer the Reader to Figure 5A-C in both these sentences.

Done, thank you.

P14, Results, I21-27: Please revise the beginning of this section to indicate that T582I and D984V were without effect on single-channel current amplitude, but G126D, I336K and T465I reduced single-channel current amplitude and refer the Reader to Figures 4 and S1.

Done, thank you.

P15, Discussion, I3-5: I336K was studied by Han ST et al. JCI Insight. 2018 Jul 26;3(14):e121159. These authors showed that it had a severe maturation defect when expressed in CF bronchial epithelial cells with CFTR function increased around 10 fold with lumacaftor and ivacaftor in Ussing chamber studies. These authors also showed that residue I336 faced away from the channel pore.

We now mention this study and its findings (P17).

P15, Discussion, I20-21: Please refer the Reader to Figure 5D and E for PKA activation of I336K and T465I.

Done, thank you.

P17, Discussion, I16-17: Please provide a reference for people with CF and G126D and I336K now being eligible for treatment with ETI. A web address will suffice.

Done, thank you.

P18, Figure legends, I2-3: Please indicate that PDB id: 6MSM is human CFTR.

Done, thank you.

P18, Figure legends, I5-6: Please refer to panels B1 - B6.

Done, thank you.

P20, Figure legends, I11-12: For the reference to ~20-fold stimulation by drugs, don't you mean Fig. 5F, 4th panel?

Corrected, thank you.

P20, Figure legends, I19: "Fig. 5F, 4th panel".

Corrected, thank you.

Figure 3A-D and F-I: Please revise the Figure Legend to indicate that downward deflections correspond to channel openings.

Done, thank you.

Figure 4A: (i) The inclusion of current amplitude scale bars for wild-type and each of the three mutants is potentially confusing. It would be preferable to have one scale bar for current amplitude as for the single scale bar for time (see also Fig. S1A). Is the G126D recording drawn to the same scale as wild-type, T582I and D984V? (ii) Please indicate the closed channel level.

The currents are shown at the same scale for all constructs. We now show a single scale bar, and have marked the zero-current levels with short black dashes.

Figure 5A-E: Please revise the Figure Legend to indicate that downward deflections correspond to channel openings.

Done, thank you.

Figure 6A-D: Please revise the Figure Legend to indicate that the continuous and dashed grey lines indicate the closed and open channel levels of wild-type CFTR.

Done, thank you.

Figure 6E: Please revise the straight line fits so that they do not extend beyond +40 mV.
Done, thank you.

Referee #2:

In the manuscript titled ‘Molecular and pharmacological evaluation of rare, cystic fibrosis causing missense mutations of the CFTR channel’, the authors sort to characterize the molecular and pharmacological effects of 5 rare CF-causing mutations. They used western blot to show that all five mutations decrease the quantity of mature CFTR protein which can be corrected by CFTR corrector VX-445 and VX-661. Micro- and macroscopic recordings are shown to demonstrate their gating and conductance defects. Potentiators VX-770 and VX-445 are shown to rescue the gating defects found which provides ‘ADDITIONAL’ evidence that ETI treatment can be used for patients carrying these CF-causing mutations. ETI has been approved as a treatment for patients carrying G126D and I336K by US-FDA about one year ago, which means Vertex Pharmaceuticals and US-FDA already had the data that point to its effectiveness on these mutants. This reduces the significance/novelty of the data shown here. However, given that those data are not publicly available, current data can still benefit the CF/CFTR community, especially there are three other mutants involved here.

The G126D and I336K mutations indeed appeared on the FDA-approved list last year, while we were finalizing the current study. However, we agree with the Reviewer that the data we present here even on those two mutants are interesting for the scientific community for the following reasons: (i) the Vertex screens are typically high-throughput screens (e.g., comparison of basal and drug-stimulated short-circuit currents) which provide no detailed mechanistic information on channel pathologies, and (ii) even that information has not been released to the public.

Here are the comments that can help the authors to revise their manuscript:

1. As abovementioned earlier approval, the author may want to revise what they mention in the second paragraph on page 4 that goes ‘However, for many rare variants, ETI ...’ to make their study more justified, although they seem to have realized this in the last paragraph of the manuscript.

We believe that that introductory sentence ("*However, for many rare variants, ETI treatment has not yet been approved. Characterizing the phenotypes and drug responses of such rare mutations is therefore a timely problem.*") is valid, as it was formulated in general, not in connection with the five specific mutations studies here.

2. In the second paragraph of ‘Results’, the authors claimed that T582 side chain forms a H-bond with the side chain of D579 and cited Zhang et al., 2018. However, upon examination of the Cryo-EM map and model from that publication, side chain information is missing for D579 in the Cryo-EM map, which indicates the assignment of the side chain orientation is not accurate, that further means this datum does not really support such claim. Nonetheless, in another map published by Gao et al., 2024, the E1371Q-CFTR map clear shows side chain density of both residues, which readily support the author’s claim. See below the two figures for comparison. The map published by Gao et al. is a better one in terms of supporting the author’s claim here and hence the authors should cite this one here, instead of Zhang et al., 2018.

We agree only partially with the Reviewer, as at lower contour levels densities for both referenced side chains can be seen even in the structure by Zhang et al. Nevertheless, we have added the Gao et al. study to better support this claim.

3. In the same paragraph, the authors claimed in the model 5UAK, 582-579 and 984-289 move further apart from each other and don't form the side interactions. Upon examination of the map and model, in that Cryo-EM map, densities for 582 and 579 are both missing, so no one really knows whether they form a salt bridge or not. Actually, the distance shows they may form a salt bridge, if the authors choose to believe the model. Similarly, in that map, while density for 984 side chain is bad, density for 289 side chain is well defined, meaning they still can possibly form a salt bridge. See the figures below. Therefore, the claims the authors made here are based on incomplete information and need to be revised to reflect rigor of interpretation of the data.

We did not intend to make strong claims based on the structures, as the current study did not investigate such structural interactions. Correspondingly, we started this paragraph (P12) with the cautious sentence *"inspection of these residues provides some hints for their potential involvement..."* Nevertheless, the Reviewer is right that our discussion may have been misleading. What we meant by "moving further apart" in 5UAK is that the alpha carbon distances increase for both pairs. Finally, instead of saying that "the interactions may not form" (in 5UAK), it would have been more appropriate to say that they "may weaken or not form" in closed channels. We have revised this sentence as follows (P12): *"Of note, in the structure of dephosphorylated ATP-free closed human CFTR (PDBID: 5uak; Liu et al., 2017)) side-chain densities for D579, T582, and D984 are absent, but for both position pairs 582-579 and 984-289 the alpha carbons move further apart from each other, suggesting that the above side chain interactions may weaken or not form in closed channels."*

Finally, we now also mention the fact that position 579 is also a CF locus, which is in line with the above predictions.

4. In figure 2C, the WT has two grey bars, is the right side one with or without correctors? If with correctors, change the color to dark grey, if not, then it is not meaningful to put it side by side there. This way gives readers the impression that correctors don't work on WT CFTR.

In Fig. 2C (and 2D) each construct is represented by two bars, the left bar plotting the data from panel A, the right bar those from panel B. For WT CFTR both bars are light gray because in both panels A and B untreated WT CFTR was used as a control. This bar – which may appear as a duplication as it was normalized to itself – appears twice in Fig. 5C, to emphasize that the Band C densities of the mutants were always normalized to the untreated WT sample detected on the same blot. (Of note, in Fig. 5D the two WT bars are not identical, as in this case the fractional contribution of Band C to the total CFTR density is plotted.) This is now better explained by the figure legend and the color keys above the plots.

Also why are both SD and SEM shown?

SEM has now been omitted.

5. Figure 5A does not show VX-770 increase channel activity for G126D, but the authors claim so in the text. No steady state is seen in this recording too. They need to either revise their interpretation of their data or re-examine their data.

We agree that the trace shown was not representative for the VX-770 response (it was the trace that provided the lowest gray symbol for the 1st panel of Fig. 5F). We have replaced the recording in Fig. 5A with a more representative sample trace.

In the same figure, I recommend the authors to change the abscissa of the table to better reflect each condition – e.g., use VX-770 + VX-445, instead of only VX-445.

Done, thank you.

6. As these data intend to provide evidence to justify prescription of ETI for patients with the mutations studied here, the reviewer is wondering why all potentiator experiments are done with pre-phosphorylated CFTR variants, which does not reflect real case where in vivo CFTR are usually phosphorylated and have ATP around. The authors need to repeat experiments in Figure 5 with phosphorylated CFTR, data from which will help to better serve the purposes of this study.

All the experiments in Figs. 3F-I, 4, 5, 6, and 7 were done on phosphorylated CFTR channels. The Reviewer must have misinterpreted the term "pre-phosphorylated". Pre-phosphorylated CFTR means CFTR phosphorylated prior to the experimental segments shown, as was explained in the Methods (P8): "*Each recording started with the following protocol: a 40-60 s baseline was recorded, then 2 mM ATP was added for 40-60 s, followed by a 1-3 minute exposure to 300 nM PKA to phosphorylate and fully activate the channels.*" We now specifically state this in each of the legends for Figures 3F-I, 4A, 5A-E, and 6A.

7. On page 16, G126D is at the periphery of the molecule, rather than in the pore region, how can this mutation affect the pore region of the channel?

We do not know. But we do cite a recent computational study (Zeng et al., 2024) which showed that G126 is located at a helical kink in TM2 which, based on molecular dynamics simulations, changes conformation during gating (P12). A mutation-induced change in the backbone conformation of TM2 might potentially impact the precise conformation of pore-lining TM1. However, we prefer to refrain from such speculations.

8. On page 17, the whole cell current prediction seems too simplified, and the actual significance of this prediction is not clear. If the authors want to make these specific claims, they need to do whole cell experiments to verify they are true or not.

We have provided a detailed account of the limitations associated with our whole-cell predictions (P19): "*One limitation of these predictions is that the above biophysical parameters were quantitated in heterologous expression systems, whereas patient-specific genetic and environmental factors may impact CFTR expression and function in vivo (Cutting, 2015). A further limitation of the predicted rank order is that these rare mutations are essentially never found in homozygous form. Thus, the severity of disease symptoms will also depend on the functional consequences of the mutation present on the other allele. The above rank order is expected only in cases where the above mutations are found in trans with a null allele, but published reports on clinical presentations for such patients are scarce.*"

9. On page 12, is it Figure 4B, right, instead of figure 2B, right, in the bottom line? In the following paragraph, I suppose there is a typo: 'for', not 'or' before

Corrected, thank you.

Dear Dr Csanády,

Re: JP-RP-2025-288955R1 "Molecular and pharmacological evaluation of rare, cystic fibrosis causing missense mutations of the CFTR channel" by Olivér Závoti and László Csanády

Thank you for submitting your manuscript to The Journal of Physiology. It has been assessed by a Reviewing Editor and by 2 expert referees and we are pleased to tell you that it is acceptable for publication following satisfactory revision.

REVISION CHECKLIST:

We look forward to receiving your revised submission.

Yours sincerely,

Peying Fong
Senior Editor
The Journal of Physiology

EDITOR COMMENTS

Senior Editor:

Thank you for your responsiveness in addressing points raised in the initial review of your manuscript. In the attached reports of your revised manuscript, both original Expert Referees express satisfaction with how you incorporated their feedback.

The Reviewing Editor indicates overall suitability for acceptance. We do note that Referee 1 suggested a few remaining corrections to improve clarity of phrasing and presentation. These instances occur within the body of the manuscript and two figure legends. A few inaccuracies in referencing also require your attention.

I anticipate that you will be able to address these final details readily.

Thank you for submitting your work to The Journal of Physiology. I look forward to receiving the final version of your manuscript soon.

REFEREE COMMENTS

Referee #1:

The authors have addressed in full previous comments on their manuscript. I offer the following minor suggestions based on reading their revised manuscript, which they may wish to consider:

Page 5, l87-89: To avoid repetition of "it", this sentence could be revised as "... upon phosphorylation the R domain is released from its inhibitory position ...".

Page 6, l124: There is an error with an author's name ("des Georges et al., 2004").

Page 13, l264: "The five CF-causing mutations are scattered throughout the channel structure".

Page 15, l334-337: "macroscopic currents" rather than "macroscopic patches".

Page 23, l534-536: "O'Riordan, C. R.".

Page 24, l584-587: "JCI Insight. 3, e121159."

Page 25, l604-607: "Eur Respir J. 57, 2002774."

Page 26, l649-651: The list of authors is incomplete "... Chou, J. L., Drumm, M. L., Iannuzzi, M. C., Collins, F. S., & Tsui, L.-C."

Page 30, l711-713: "The positions of the five CF-causing mutations mapped ...".

Pages, 30-31, Figure 2 legend and Figure 3 legend: Please consider indicating what the asterisks used to denote statistical significance in Figures 2 and 3 represent in these Figure legends.

Figures 4A, 6A and 7A - E: In Figure 4A, the current amplitude scale bar is a negative value, whereas in Figures 6A and 7A - E, these scale bars are positive values. Please be consistent.

Referee #2:

The authors have addressed all my comments.

END OF COMMENTS

RESPONSES TO REFEREE COMMENTS

Referee #1:

The authors have addressed in full previous comments on their manuscript. I offer the following minor suggestions based on reading their revised manuscript, which they may wish to consider:

We thank the Reviewer for the meticulous proofing, we have amended all the the errors.

Page 5, l87-89: To avoid repetition of "it", this sentence could be revised as "... upon phosphorylation the R domain is released from its inhibitory position ...".

Corrected.

Page 6, l124: There is an error with an author's name ("des Georges et al., 2004").

Corrected.

Page 13, l264: "The five CF-causing mutations are scattered throughout the channel structure".

Corrected.

Page 15, l334-337: "macroscopic currents" rather than "macroscopic patches".

We appreciate the Reviewer's concern that "macroscopic" should refer to "currents" rather than to "patches". On the other hand, it is the patches, rather than the currents, that are exposed to ATP. We have therefore replaced "*exposures of macroscopic patches to various test ATP concentrations*" with "*exposures of patches with multiple channels to various test ATP concentrations*".

Page 23, l534-536: "O'Riordan, C. R."

Corrected.

Page 24, l584-587: "JCI Insight. 3, e121159."

Corrected.

Page 25, l604-607: "Eur Respir J. 57, 2002774."

Corrected.

Page 26, l649-651: The list of authors is incomplete "... Chou, J. L., Drumm, M. L., Iannuzzi, M. C., Collins, F. S., & Tsui, L.-C."

Corrected.

Page 30, l711-713: "The positions of the five CF-causing mutations mapped ..."

Corrected.

Pages, 30-31, Figure 2 legend and Figure 3 legend: Please consider indicating what the asterisks used to denote statistical significance in Figures 2 and 3 represent in these Figure legends.

We have added appropriate explanations.

Figures 4A, 6A and 7A - E: In Figure 4A, the current amplitude scale bar is a negative value, whereas in Figures 6A and 7A - E, these scale bars are positive values. Please be consistent.

Corrected, all scale bars are positive now.

Dear Dr Csanády,

Re: JP-RP-2025-288955R2 "Molecular and pharmacological evaluation of rare, cystic fibrosis causing missense mutations of the CFTR channel" by Olivér Závoti and László Csanády

We are pleased to tell you that your paper has been accepted for publication in The Journal of Physiology.

Yours sincerely,

Peying Fong
Senior Editor
The Journal of Physiology

If you would like to receive our 'Research Roundup', a monthly newsletter highlighting the cutting-edge research published in The Physiological Society's family of journals (The Journal of Physiology, Experimental Physiology, Physiological Reports, The Journal of Nutritional Physiology and The Journal of Precision Medicine: Health and Disease), please click this link, fill in your name and email address and select 'Research Roundup':
<https://www.physoc.org/journals-and-media/membernews>

- You can help your research get the attention it deserves! Check out Wiley's free Promotion Guide for best-practice recommendations for promoting your work at: www.wileyauthors.com/eeo/guide. You can learn more about Wiley Editing Services which offers professional video, design, and writing services to create shareable video abstracts, infographics, conference posters, lay summaries, and research news stories for your research at: www.wileyauthors.com/eeo/promotion.

EDITOR COMMENTS

Senior Editor:

I appreciate your patience with the delay experienced as final checks of your revised manuscript were performed. It is my pleasure to inform you that the Reviewing Editor and I agree the present version addresses all points raised during the last round of review satisfactorily.

Congratulations on a fine study! We thank you for your continued contributions to The Journal of Physiology.